# Antagonistic control of *Caenorhabditis elegans* germline stem cell proliferation and differentiation by PUF proteins FBF-1 and FBF-2

**Xiaobo Wang, Mary Ellenbecker, Benjamin Hickey, Nicholas J Day, Emily Osterli, Mikaya Terzo, Ekaterina Voronina***

Division of Biological Sciences, University of Montana, Missoula, United States

**Abstract** Stem cells support tissue maintenance, but the mechanisms that coordinate the rate of stem cell self-renewal with differentiation at a population level remain uncharacterized. We find that two PUF family RNA-binding proteins FBF-1 and FBF-2 have opposite effects on *Caenorhabditis elegans* germline stem cell dynamics: FBF-1 restricts the rate of meiotic entry, while FBF-2 promotes both cell division and meiotic entry rates. Antagonistic effects of FBFs are mediated by their distinct activities toward the shared set of target mRNAs, where FBF-1-mediated post-transcriptional control requires the activity of CCR4-NOT deadenylase, while FBF-2 is deadenylase-independent and might protect the targets from deadenylation. These regulatory differences depend on protein sequences outside of the conserved PUF family RNA-binding domain. We propose that the opposing FBF-1 and FBF-2 activities serve to modulate stem cell division rate simultaneously with the rate of meiotic entry.

**\*For correspondence:**
ekaterina.voronina@mso.umt.edu

**Competing interests:** The authors declare that no competing interests exist.

## Introduction

Adult tissue maintenance relies on the activity of stem cells that self-renew and produce differentiating progeny (*Morrison and Kimble, 2006*). It is essential that self-renewal be balanced with differentiation to preserve the size of the stem cell pool over time. One simple mechanism achieving this balance is an asymmetric division that always produces a single stem cell daughter and a daughter destined to differentiate (*Chen et al., 2016*). Alternatively, tissue homeostasis can be controlled at a population level (*Simons and Clevers, 2011*), where some stem cells are lost through differentiation while others proliferate, with both outcomes occurring at the same frequency. Such population-level control of stem cell activity is observed in the *Caenorhabditis elegans* germline (*Kimble and Crittenden, 2007*). However, the mechanisms of population-level control of stem cell proliferation and differentiation in the adult tissues are largely unclear.

The *C. elegans* hermaphrodite germline is a robust system to explore the mechanisms coordinating stem cell proliferation and differentiation. It is maintained by a stem cell niche that supports about 200–250 mitotically dividing stem and progenitor cells at the distal end of the gonad (collectively called SPCs, *Figure 1A,Bii*). A single somatic distal tip cell serves as a stem cell niche and activates the GLP-1/Notch signaling necessary for SPC pool maintenance (*Austin and Kimble, 1987*), which in turn supports germline development (*Hansen and Schedl, 2013*). As germline stem cells move proximally away from the niche, they differentiate by entering meiotic prophase and eventually generate gametes near the proximal gonad end. Mitotic divisions of SPCs are not oriented and there doesn't appear to be a correlation between the position of cell divisions distributed throughout the SPC zone and the position of cells undergoing meiotic entry at the proximal end of the zone

**Figure 1.** FBF-1 and FBF-2 differentially regulate the extent of germline stem and progenitor cell (SPC) zone. (**A**) Schematic of the distal germline of *C. elegans* adult hermaphrodite. In this and following images, germlines are oriented with their distal ends to the left. GLP-1/Notch signaling from the distal tip cell (blue) supports germline SPC proliferation. Progenitors enter meiosis in the transition zone. FBF-1 and FBF-2, downstream of GLP-1/Notch, are required for SPC maintenance. Green circles, stem and progenitor cells; red diamonds, mitotically dividing cells. (**B**) Distal germlines dissected from adult wild type, *fbf-1(lf)*, and *fbf-2(lf)* hermaphrodites and stained with anti-REC-8 (green) and anti-*phospho*-Histone *H3* (pH3; red) to visualize the SPC zone and mitotic cells in M-phase. Germlines are outlined with the dashed lines and the vertical dotted line marks the beginning of transition zone as recognized by the 'crescent-shaped' chromatin and loss of REC-8. Scale bar: 10 μm. (**C**) SPC zone lengths of the wild type, *fbf-1(lf)* and *fbf-2(lf)* germlines were measured by counting germ cell diameters (gcd) spanning SPC zone. Genetic background is indicated on the X-axis and the extent of SPC zone on the Y-axis. Differences in SPC zone lengths were evaluated by one-way ANOVA with Dunnett's post-test. Data were collected from three independent experiments, with 10–15 germlines per strain per replicate. (**D**) Median SPC G2-phase length in different genetic backgrounds, as indicated on the X-axis. Difference in median G2 length was evaluated by one-way ANOVA with Dunnett's post-test. G2 length was estimated in three independent experiments as shown in *Figure 1—figure supplement 1C*, each replicate experiment involved analysis of 145–159 germlines per strain. (**E**) Larval germ cell doubling time in different genetic backgrounds (as indicated on the X-axis). Plotted values are individual data points and means ± SD. Difference in germ cell doubling time was evaluated by one-way ANOVA with Dunnett's post-test. Data were collected from four independent replicates as shown in *Figure 1—figure supplement 1E,F*, each analyzing 15–21 germlines per strain per time point (144–148 germlines per strain total). (**F**) Meiotic entry rate of germline progenitors in different genetic backgrounds indicated on the X-axis. Differences in meiotic entry rate between each *fbf* and the wild type were evaluated by one-way ANOVA with T-test with Bonferroni correction post-test. Meiotic entry rates were estimated in five independent experiments as shown in *Figure 1—figure supplement 1G*, each analyzing 5–7 germlines per strain per time point (89–94 germlines per strain total). (**B–F**) All experiments were performed at 24°C. Plotted values are individual data points and means ± SD. Asterisks mark statistically significant differences (****, p<0.0001; **, p<0.01; *, p<0.05).

The online version of this article includes the following figure supplement(s) for figure 1:

**Figure supplement 1.** SPC dynamics in different genetic backgrounds.

*Figure 1 continued on next page*

*Figure 1 continued*

**Figure supplement 2.** The number of SPC cell rows is a robust estimate of progenitor numbers.

(*Crittenden et al., 2006*; *Fox et al., 2011*; *Jaramillo-Lambert et al., 2007*; *Maciejowski et al., 2006*).

Analysis of *C. elegans* germline stem cell maintenance identified a number of genes affecting SPC self-renewal and differentiation (*Hansen and Schedl, 2013*). Genes essential for self-renewal include GLP-1/Notch and two highly similar Pumilio and FBF (PUF) family RNA-binding proteins called FBF-1 and FBF-2 (*Austin and Kimble, 1987*; *Crittenden et al., 2002*; *Zhang et al., 1997*). Genetic studies of stem cell maintenance led to a model where a balance of mitosis- and meiosis-promoting activities maintains tissue homeostasis (*Hubbard and Schedl, 2019*), but the regulatory mechanism coordinating proliferative SPC activity with meiotic entry remained elusive.

Importantly, the SPC cell cycle is distinct from that of most somatic stem cells. One characteristic feature of *C. elegans* germline SPC cell cycle is a very short G1 phase (*Fox et al., 2011*; *Furuta et al., 2018*), reminiscent of the short G1 phase observed in mammalian embryonic stem cells (ESCs) (*Becker et al., 2006*; *Kareta et al., 2015*; *White and Dalton, 2005*). Mouse and human ESCs maintain robust cell division rates supported by a cell cycle with a short G1 phase, while the length of S and G2 phases is similar to that observed in differentiated somatic cells (*Becker et al., 2006*; *Chao et al., 2019*; *Kareta et al., 2015*; *Stead et al., 2002*). Despite the abbreviated G1 phase, ESCs maintain S and G2 checkpoints (*Chuykin et al., 2008*; *Stead et al., 2002*; *White and Dalton, 2005*). Similarly, *C. elegans* SPCs retain G2 checkpoints despite the shortened G1 phase (*Butuči et al., 2015*; *Garcia-Muse and Boulton, 2005*; *Lawrence et al., 2015*; *Moser et al., 2009*). This modified cell cycle may be due to a constant proliferative demand that both SPCs and ESCs are subject to. By contrast, this type of modified cell cycle is not observed in the adult stem cell populations that support regenerative response upon injury, such as adult mammalian bulge stem cells (hair follicle stem cells; *Cotsarelis et al., 1990*) or satellite cells (muscle stem cells; *Schultz, 1974*; *Schultz, 1985*; *Snow, 1977*) that remain in G0 or quiescent phase for the most of the adult life and only reenter cell cycle upon injury. Similarly, adult epidermal stem cells regulate their cell cycle by controlling the G1/S transition to maintain tissue homeostasis (*Mesa et al., 2018*).

Unlike somatic cells' G1 phase that is triggered and marked by increased amounts of cyclins E and D (*Aleem et al., 2005*; *Guevara et al., 1999*), germ cells and ESCs are characterized by a shortened G1 phase and maintain a constitutive robust expression of G1/S regulators Cyclin E and CDK2 throughout the cell cycle (*Fox et al., 2011*; *Furuta et al., 2018*; *White and Dalton, 2005*). Despite continuous proliferation of *C. elegans* SPCs, the SPC mitotic rate changes during development and in different mutant backgrounds (*Kocsisova et al., 2019*; *Michaelson et al., 2010*; *Roy et al., 2016*), and it is unknown how SPC division and meiotic entry rates might be altered while maintaining the cell cycle with an abbreviated G1 phase. Here, we report the mechanism through which PUF family RNA binding proteins FBF-1 and FBF-2 simultaneously change the rates of SPC cell cycle progression and meiotic entry.

PUF proteins are expressed in germ cells of many animals and are conserved regulators of stem cells (*Salvetti, 2005*; *Wickens et al., 2002*). *C. elegans* PUF proteins expressed in germline SPCs, FBF-1 and FBF-2, share the majority of their target mRNAs (*Porter et al., 2019*; *Prasad et al., 2016*) and are redundantly required for SPC maintenance (*Crittenden et al., 2002*; *Zhang et al., 1997*). Despite 89% identity between FBF-1 and FBF-2 protein sequences, several reports suggest that FBF-1 and FBF-2 localize to distinct cytoplasmic RNA granules and have unique effects on the germline SPC pool (*Lamont et al., 2004*; *Voronina et al., 2012*). FBF-1 and FBF-2 each support distinct numbers of SPCs (*Lamont et al., 2004*). Furthermore, FBF-1 inhibits accumulation of target mRNAs in SPCs, while FBF-2 primarily represses translation of the target mRNAs (*Voronina et al., 2012*). Some differences between FBF-1 and FBF-2 function might be explained by their association with distinct protein cofactors, as we previously found that a small protein DLC-1 is a cofactor specific to FBF-2 that promotes FBF-2 localization and function (*Wang et al., 2016*). Despite the fact that several repressive mechanisms have been documented for PUF family proteins (*Quenault et al., 2011*), it is relatively understudied how the differences between PUF homologs are specified. Here, we sought to take advantage of the distinct SPC numbers maintained by individual FBF proteins to

understand how they regulate the dynamics of SPCs cell cycle and meiotic entry and to probe the functional differences between FBFs.

Elaborating on the general contribution of PUF proteins to stem cell maintenance, we describe here that FBF-1 and FBF-2 have antagonistic effects on the rates of germline SPC cell cycle and meiotic entry. We find that FBFs regulate core cell cycle machinery transcripts along with transcripts required for differentiation to coordinately affect both transcript classes. FBF-1 requires CCR4-NOT deadenylation machinery, while FBF-2 functions independently of CCR4-NOT and might protect target mRNAs from deadenylation. These distinct functions of FBFs are determined by the protein regions outside of the conserved PUF homology domain. The opposing regulation of SPC cell cycle and differentiation by FBFs allows stem cells to simultaneously modulate cell division rate and meiotic entry.

## Results

### FBF-1 and FBF-2 differentially modulate cell division and meiotic entry of *C. elegans* germline SPCs

During tissue maintenance, stem cells adjust their proliferative activity and differentiation rate through diverse regulatory mechanisms, including RNA-binding protein-mediated post-transcriptional regulation. We hypothesized that two paralogous RNA-binding proteins FBF-1 and FBF-2 differentially regulate germline stem cell mitotic rate and meiotic entry in *C. elegans*, resulting in distinct effects on the size of stem and progenitor cell (SPC) zone. We first determined how the extent of SPC zone was affected by loss-of-function mutations of each *fbf*. SPCs were marked by staining for the nucleoplasmic marker REC-8 (*Figure 1A,B*; *Hansen et al., 2004*), and the extent of SPC zone was measured by counting the number of cell rows positive for REC-8 staining in each germline. Consistent with a previous report (*Lamont et al., 2004*), we observed that the SPC zone of *fbf-1(ok91, loss-of-function mutation, lf)* (~15 germ cell diameters, gcd; *Figure 1Bi*) is smaller than that of the wild type (~20 gcd, *Figure 1Bii*), whereas the SPC zone of *fbf-2(q738, loss-of-function mutation, lf)* (~25 gcd, *Figure 1Biii*) is larger than that of the wild type (*Figure 1B,C*). The differences in the length of SPC zone between *fbf* single mutants and the wild type are consistently observed in animals through the first day of adulthood (*Figure 1—figure supplement 1A*).

To test whether the differences in the lengths of germline SPC zone between *fbf* mutants and the wild type result from changes in the rate of cell division, we compared cell cycle parameters in each genetic background. We started with measuring the M-phase index (the percentage of SPC zone cells in M phase) following immunostaining for the SPC marker REC-8 and the M-phase marker phospho-histone H3 (pH3, *Figure 1B*). We found that the mitotic index of *fbf-1(lf)* was significantly higher than that of the wild type (by 54%, *Figure 1—figure supplement 1B*). By contrast, the mitotic index of *fbf-2(lf)* was significantly lower than that of the wild type (by 42%; *Figure 1—figure supplement 1B*). These results suggested that loss of FBF-2 might reduce SPC proliferation. We also considered the possibility that the loss of FBF-1 might accelerate progression of SPCs through the cell cycle. However, as described below, this hypothesis was rejected. Since *C. elegans* germline stem cells have abbreviated G1 and extended G2 phases (*Fox et al., 2011*), we tested whether the G2-phase duration is affected differentially by loss of function mutation of each *fbf*. Using phospho-histone H3 immunostaining and 5-ethynyl-2′-deoxyuridine (EdU) pulse, we estimated a median G2 length by determining when 50% of pH3-positive cells become EdU-positive (*Figure 1—figure supplement 1C*). We found that the median G2 length of *fbf-2(lf)* is significantly greater than that of the wild type, suggesting that loss of FBF-2 results in slower progression through the G2-phase of the cell cycle (by 25%; *Figure 1D*). By contrast, the median G2 length of *fbf-1(lf)* is not significantly different from that of the wild type (*Figure 1D*). We conclude that FBF-2 accelerates SPC cell cycle by facilitating the G2-phase progression.

Since mutation of *fbf-1* did not affect the length of G2 phase, we tested whether percentage of SPCs in S phase is affected by this mutation. We determined percent SPCs labeled by EdU during a 30-min pulse (*Fox et al., 2011*) and found a minor increase in S-phase index in *fbf-1(lf)* compared to the wild type (*Figure 1—figure supplement 1D*). These results refute the interpretation that *fbf-1(lf)* mutation causes faster cell cycle progression.

To directly estimate the rate of germ cell division in wild type and *fbf* mutants, we assayed germ cell proliferation during larval development before the onset of meiotic differentiation. In *C. elegans*, two primordial germ cells in L1 larvae proliferate to produce germline stem cell pools of 20–30 cells in L2 larval stage within 20 hr (*Hansen et al., 2004*; *Hirsh et al., 1976*; *Pepper et al., 2003b*; *Figure 1—figure supplement 1E*). We found that *fbf-1(lf)* did not affect the rate of germ cell division, while *fbf-2(lf)* dramatically reduced germ cell accumulation (*Figure 1—figure supplement 1F*). Exponential fits revealed that *fbf-2(lf)* significantly increased SPC doubling time from 6.1 hr to 8.1 hr (*Figure 1E*). By contrast, there was no significant difference in germ cell proliferation rate between *fbf-1(lf)* and the wild type. We conclude that the cell division rate is decreased in *fbf-2(lf)* and unaffected in *fbf-1(lf)*.

Despite the same SPC cell division rate, the SPC zone of *fbf-1(lf)* is smaller than that of the wild type, suggesting a possibility that *fbf-1(lf)* might result in faster meiotic entry. Conversely, compared to the wild type, *fbf-2(lf)* maintains a relatively larger SPC population but with slower proliferation, suggesting that the rate of meiotic entry in *fbf-2(lf)* might be slower than in the wild type. To test these possibilities, we determined the rate of meiotic entry in each genetic background. Animals were continuously EdU labeled and stained for EdU and REC-8 at three time points. The number of germ cells negative for REC-8 but positive for EdU were scored at each time point and the rate of meiotic entry was estimated from the slope of the plotted regression line as in *Figure 1—figure supplement 1G*. We found that *fbf-1(lf)* results in a significantly increased rate of meiotic entry compared to the wild type (by 31%; *Figure 1F*), whereas *fbf-2(lf)* results in a significantly reduced rate of meiotic entry (by 18%; *Figure 1F*). We conclude that FBF-2 stimulates meiotic entry while FBF-1 restricts meiotic entry.

Slow germ cell proliferation together with a delay in meiotic entry enhance the penetrance of germline tumor formation in sensitized genetic backgrounds (*Hubbard and Schedl, 2019*; *Killian and Hubbard, 2005*). We tested whether *fbf-2(lf)* enhances the overproliferative phenotype of the weak *glp-1* gain-of-function allele, *glp-1(ar202)*. We found that *fbf-2(lf)* is a strong enhancer of *glp-1(gf)* since 97% *fbf-2(lf); glp-1(gf)* animals have tumorous germlines with 24% germlines showing complete tumors, even at the permissive temperature of 15°C (*Table 1*).

In summary, mutations in *fbf-1* and *fbf-2* differentially influence both SPC cell cycle and meiotic entry rate, suggesting FBF proteins have antagonistic effects on SPC proliferation and differentiation. FBF-1 promotes a more quiescent stem cell state characterized by a slower rate of meiotic entry, while FBF-2 promotes a more activated stem cell state characterized by faster rates of both cell cycle and meiotic entry. While FBF's effects on transcripts regulating meiotic entry such as *gld-1* have been documented previously (*Crittenden et al., 2002*; *Brenner and Schedl, 2016*), FBF regulation of the cell cycle has been unexplored. We hypothesized that FBFs control SPC progression through cell cycle by regulating B-type cyclin mRNAs.

**Table 1.** *fbf-2(lf)* enhances the overproliferation phenotype of *glp-1(gf)* at 15°C.

All animals were maintained at 15°C. For each genotype, after scoring sterility as the lack of embryos in the uterus, germlines of sterile animals were dissected and stained with DAPI, anti-REC-8 antibodies, and anti-phospho-histone H3 antibodies for evaluation of overproliferation. All animals were analyzed at 1 day after L4 stage. Tum, a complete tumorous germline. Pro, proximal overproliferation phenotype.

| Genotype | Normal germline, % | Tum, % | Pro, % | N |
|---|---|---|---|---|
| *fbf-2(q738lf)* | 100 | 0 | 0 | *many* |
| *fbf-1(ok91lf)* | 100 | 0 | 0 | *many* |
| *glp-1(ar202gf)* | 100 | 0 | 0 | 231 |
| *fbf-2(lf); glp-1(gf)* | 3 | 24 | 74 | 104 |
| *fbf-1(lf); glp-1(gf)* | 99 | 0 | 1 | 414 |

## PUF-mediated repression of cyclin B limits accumulation of germline SPCs in *fbf-2*

Cyclin B/Cdk1 kinase, also known as M-phase promoting factor, triggers G2/M transition in most eukaryotes (*Lindqvist et al., 2009*). Four cyclin B family genes provide overlapping as well as specific mitotic functions in *C. elegans* (*van der Voet et al., 2009*). We hypothesized that the slower G2-phase and lower M-phase index of *fbf-2(lf)* SPCs results from translational repression and reduced steady-state levels of four cyclin B family transcripts mediated by the remaining germline-expressed PUF-family proteins, including FBF-1. We addressed this hypothesis in two ways. First, we tested whether mutation of FBF-binding elements (FBEs) in the 3'UTR of *cyb-2.1* mRNA would result in translational derepression of *cyb-2.1*. Second, we assessed whether derepression of *cyb-2.1* in *fbf-2 (lf)* would lead to accumulation of more SPCs by uncoupling PUF-mediated regulation of cell division and meiotic entry.

FBFs repress their target mRNAs by binding to the F̲BF-b̲inding e̲lements (FBEs; UGUxxxAU) in the 3'UTRs (*Bernstein et al., 2005*; *Crittenden et al., 2002*; *Merritt and Seydoux, 2010*). Four mRNAs encoding Cyclin B family members co-purify with FBF proteins and three of them contain predicted FBEs in their 3'UTRs (*Porter et al., 2019*; *Prasad et al., 2016*). Since *cyb-2.1* mRNA is found in complex with FBFs across multiple experimental conditions and contains more canonical FBE sites in its 3'UTR than the other cyclin B transcripts, we selected *cyb-2.1* for further analysis. If FBFs repress translation of *cyb-2.1* by binding to FBEs, mutation of FBEs would cause derepression of CYB-2.1 protein. To test this prediction, we established a transgenic animal *3xflag::cyb-2.1(fbm)*, expressing 3xFLAG::CYB-2.1 under the control of 3'UTR with mutated FBEs (ACAxxxAU); as a control, a transgenic animal expressing *3xflag::cyb-2.1(wt)* with wild-type FBEs was also established (*Figure 2A*). Quantification of transgene transcript levels by qPCR suggested that steady-state transcript levels of *3xflag::cyb-2.1(fbm)* were ~4.5 fold greater than those of *3xflag::cyb-2.1(wt)*, suggesting that FBEs affect steady-state transcript levels (*Figure 2B*). By immunoblotting, we found that the expression of 3xFLAG::CYB-2.1 protein was increased ~1.4 fold in *3xflag::cyb-2.1(fbm)* animals compared to *3xflag::cyb-2.1(wt)*, suggesting that the presence of FBEs decreases protein production from *cyb-2.1* mRNA (*Figure 2C*). The abundance of cyclin family proteins is subject to extensive post-translational control (*Langenfeld et al., 1997*; *Peters, 2002*), which might account for a larger difference observed at the level of transcript. *C. elegans* SPCs express five PUF-family proteins that cluster into three groups based on sequence similarity: FBF-1/–2, PUF-8, and PUF-3/–11 (*Ariz et al., 2009*; *Crittenden et al., 2002*; *Haupt et al., 2020*; *Lamont et al., 2004*; *Stumpf et al., 2008*). Each of the three PUF groups has a distinct RNA-binding specificity (*Bernstein et al., 2005*; *Koh et al., 2009*; *Opperman et al., 2005*), so it is likely that FBEs in *cyb-2.1* 3'UTR are predominantly recognized and regulated by FBFs. However, we cannot exclude the possibility that FBEs in the *cyb-2.1* 3'UTR mediate association with PUFs other than FBF-1/–2. We conclude that *cyb-2.1* expression in SPCs is downregulated by PUF proteins recruited to the FBEs.

Loss-of-function mutation in *fbf-2* is associated with slower SPC cell division in conjunction with a slower SPC meiotic entry rate. We hypothesized that both these phenotypes might be mediated by reduced translation of key FBF target mRNAs that are required for cell cycle progression or meiotic entry respectively. We aimed to disrupt coordinate repression of cell cycle- and differentiation-related transcripts in *fbf-2(lf)* by introducing *3xflag::cyb-2.1(fbm)* transgene that produces increased levels of corresponding mRNA and protein. *fbf-2(lf)* with its slow cell division rate provides a sensitized background for testing the effects of cyclin B deregulation on cell cycle dynamics since it is not clear whether SPC cell cycle rate could be accelerated beyond that of the wild type. We hypothesized that the slower SPC cell cycle in *fbf-2(lf)* is caused by PUF-mediated repression of cyclin B-family mRNAs. If any cyclin B-family gene can promote SPC proliferation, disrupting translational repression of a single cyclin B-family transcript in *fbf-2(lf)* would rescue the slow cell cycle phenotype and accelerate SPC cell division. To test this hypothesis, we estimated the doubling time of larval germ cells after crossing the *3xflag::cyb-2.1fbm* and *3xflag::cyb-2.1wt* transgenes into *fbf-2(lf)* genetic background. We found that the SPC doubling time of *fbf-2(lf); 3xflag::cyb-2.1fbm* was significantly shorter than that of *fbf-2(lf)* (*Figure 2D*). By contrast, there was no significant difference in the doubling time between *fbf-2(lf); 3xflag::cyb-2.1wt* and *fbf-2(lf)* (*Figure 2D*). We expected that overexpression of 3xFLAG::CYB-2.1 in *fbf-2(lf)* genetic background would not affect SPC meiotic

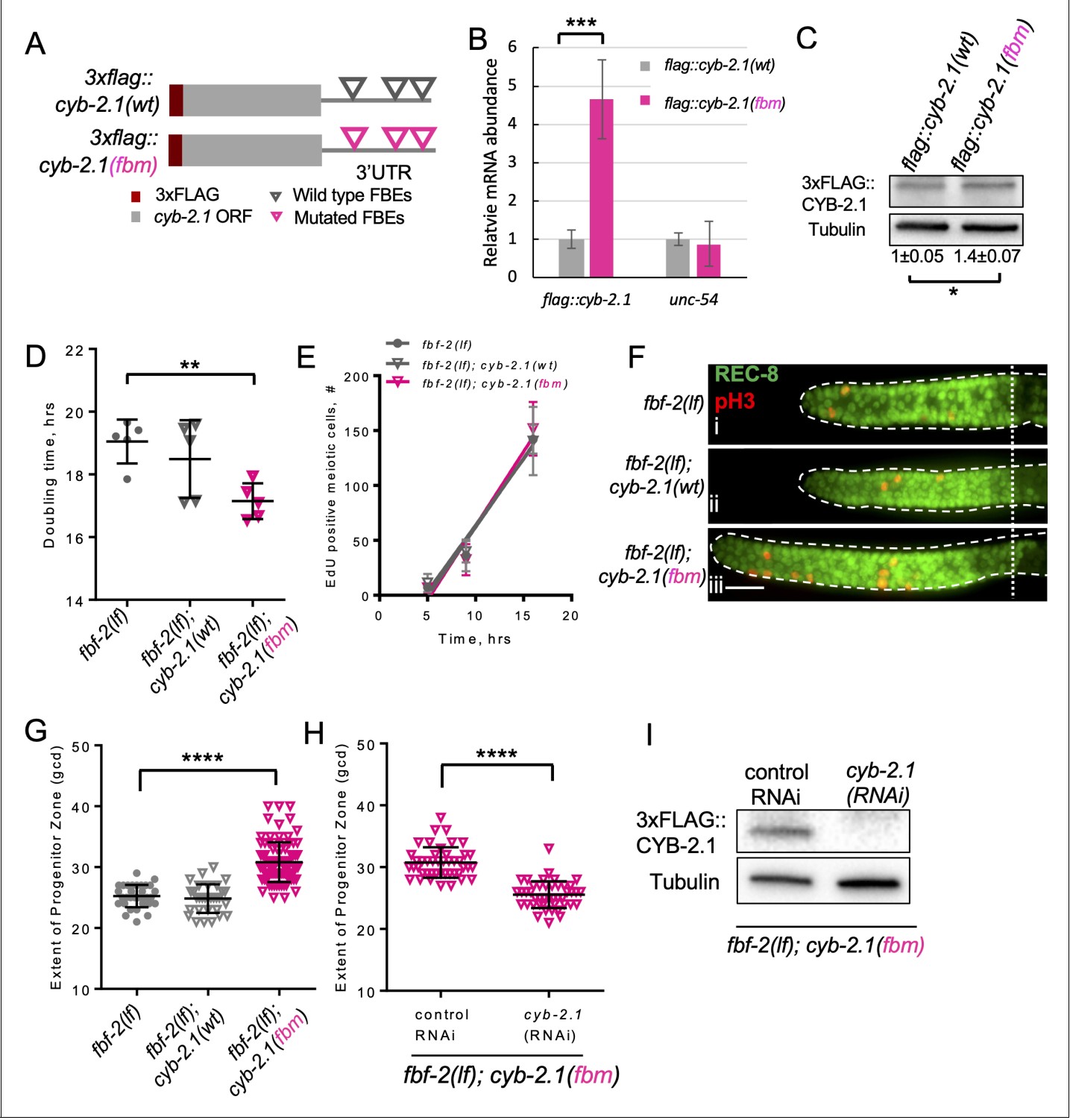

**Figure 2.** FBF-mediated repression of cyclin B limits accumulation of germline progenitor cells. (**A**) Schematic representation of transgenes encoding 3xFLAG-tagged CYB-2.1(wt) with wild type FBF binding elements (FBEs, UGUxxxAU) in 3'UTR and 3xFLAG-tagged CYB-2.1(fbm) with *F*BF *b*inding elements *m*utated (ACAxxxAU). (**B**) qRT-PCR of *3xflag::cyb-2.1* and *unc-54* transcripts in *3xflag::cyb-2.1(wt)* and *3xflag::cyb-2.1(fbm)* worms using actin (*act-1*) as a normalization control. Data represent two replicates, values are average ± SD. Differences in levels were evaluated by T-test; asterisks mark statistically significant difference (p<0.001). (**C**) Immunoblot analysis of 3xFLAG::CYB-2.1 protein levels in *3xflag::cyb-2.1(wt)* and *3xflag::cyb-2.1(fbm)* worms using α-tubulin as a loading control. Data represent three replicates, values are average ± SD. Differences in levels were evaluated by T-test; asterisk marks statistically significant difference (p<0.05). (**D**) Larval germ cell doubling time in different genetic backgrounds (as indicated on the

*Figure 2 continued on next page*

Figure 2 continued

X-axis). Plotted values are individual data points and means ± SD. Difference in germ cell doubling time was evaluated by one-way ANOVA with Dunnett's post-test. (E) Meiotic entry of progenitors in different genetic backgrounds. Time course of accumulating EdU-labeled, REC-8 negative germ cells in different genetic backgrounds in one biological replicate (the data are representative of two biological replicates, each analyzing 6–7 germlines per strain per time point, up to 41–42 germlines per strain total). X-axis displays time points when animals were dissected for staining for EdU and REC-8. Y-axis indicates the number of EdU-positive cells that are negative for REC-8. Plotted values are means ± SD. (F) Distal germlines dissected from the *fbf-2(lf)*, *fbf-2(lf); cyb-2.1(fbm)* and *fbf-2(lf); cyb-2.1(wt)* animals and stained with anti-REC-8 (green) and anti-pH3 (red). Germlines are outlined with dashed lines and the vertical dotted line marks the beginning of transition zone. Scale bar: 10 μm. (G) The extent of SPC zone in the *fbf-2(lf)*, *fbf-2(lf); cyb-2.1(fbm)* and *fbf-2(lf); cyb-2.1(wt)* genetic backgrounds. Plotted values are individual data points and means ± SD. Differences in SPC zone lengths were evaluated by one-way ANOVA with Dunnett's post-test; asterisks mark statistically significant difference (p<0.0001). Data were collected from two independent experiments and 14–19 germlines were scored for each genotype per replicate. (H) The extent of SPC zone in the *fbf-2(lf); cyb-2.1(fbm)* after *cyb-2.1(RNAi)* compared to the empty vector RNAi control. Plotted values are individual data points and means ± SD. Differences in SPC zone lengths were evaluated by T-test; asterisks mark statistically-significant difference (p<0.0001). Data were collected from two independent experiments and 44 independent germlines were scored for each condition. (I) Immunoblot analysis of 3xFLAG::CYB-2.1 protein levels in *3xflag::cyb-2.1fbm* after *cyb-2.1(RNAi)* compared to the empty vector RNAi control. Tubulin was used as a loading control. (B–I) All experiments were performed at 15°C.

entry rate. Indeed, there was no significant difference in SPC meiotic entry among *fbf-2(lf)*, *fbf-2(lf); cyb-2.1wt*, and *fbf-2(lf); cyb-2.1(fbm)* (*Figure 2E*).

Accelerated SPC cell cycle without a change in SPC meiotic entry rate is expected to result in accumulation of SPCs and an increase of SPC zone length. To test this prediction, we measured the extent of SPC zone of *fbf-2(lf); 3xflag::cyb-2.1fbm* and *fbf-2(lf); 3xflag::cyb-2.1wt*. We found that the SPC zone of *fbf-2(lf); 3xflag::cyb-2.1fbm* (~32 gcd, *Figure 2Fiii*) is significantly larger than that of the *fbf-2(lf)* (~26 gcd, *Figure 2Fi*, G, p<0.0001). By contrast, there is no significant difference in the length of SPC zone between the *fbf-2(lf); 3xflag::cyb-2.1wt* and *fbf-2(lf)* (*Figure 2Fii*, G). To test whether the expansion of SPC zone in *fbf-2(lf); 3xflag::cyb-2.1fbm* results from overexpression of *cyb-2.1*, we measured the extent of SPC zone following knockdown of *cyb-2.1* by RNAi. We found that the SPC zone of *fbf-2(lf); 3xflag::cyb-2.1fbm* after *cyb-2.1(RNAi)* became significantly shorter (~26 gcd) compared to the control RNAi (~31 gcd; *Figure 2H*). Depletion of CYB-2.1 was confirmed by immunoblot for FLAG::CYB-2.1 after RNAi of *cyb-2.1* compared to the control (*Figure 2I*).

We conclude that disrupting PUF-mediated regulation of CYB-2.1 has uncoupled cell cycle dynamics from the rate of meiotic entry in the *fbf-2(lf)* background, supporting the model of coordinated regulation of cell cycle and meiotic entry by FBFs. These results further suggest that meiotic entry rate and cell cycle progression are regulated through distinct subsets of FBF targets, rather than meiotic entry rate being a direct consequence of how fast SPCs are generated by cell divisions.

## FBF-1 function requires CCR4-NOT deadenylase complex

One mechanism of PUF-dependent destabilization of target mRNAs is through recruitment of CCR4-NOT deadenylase that shortens poly(A) tails of the targets (*Quenault et al., 2011*). CCR4-NOT deadenylase is a complex that includes three core subunits: two catalytic subunits CCR-4/CNOT6/6L and CCF-1/CNOT-7/8 and a scaffold subunit LET-711/CNOT1, which are highly conserved in *C. elegans* and humans (*Figure 3A*; *Nousch et al., 2013*). Although multiple PUF family proteins, including FBF homologs in *C. elegans*, interact with a catalytic subunit of CCR4-NOT in vitro, the contribution of CCR4-NOT to PUF-mediated repression in vivo is still controversial (*Suh et al., 2009*; *Weidmann et al., 2014*). We hypothesized that the enlarged germline SPC zone in *fbf-2(lf)* mutant results from FBF-1-mediated destabilization and translational repression of target mRNAs required for meiotic entry achieved through the activity of CCR4-NOT deadenylase. If so, knockdown of CCR4-NOT in *fbf-2(lf)* genetic background would lead to derepression of target mRNAs in SPCs and a decrease of the length of SPC zone.

First, we measured the extent of SPC zone after RNAi-mediated knockdown of core CCR4-NOT subunits, and found that CCR4-NOT RNAi dramatically shortened the SPC zone in *fbf-2(lf)* compared to the control RNAi (p<0.01; *Figure 3B*). By contrast, the lengths of SPC zones in the wild type and *fbf-1(lf)* animals were not significantly affected by CCR4-NOT knockdown (*Figure 3B*). We note that the observed effects of CCR4-NOT knockdown are milder than those reported by a recent publication *Nousch et al., 2019*; these differences might result from the later developmental stage affected

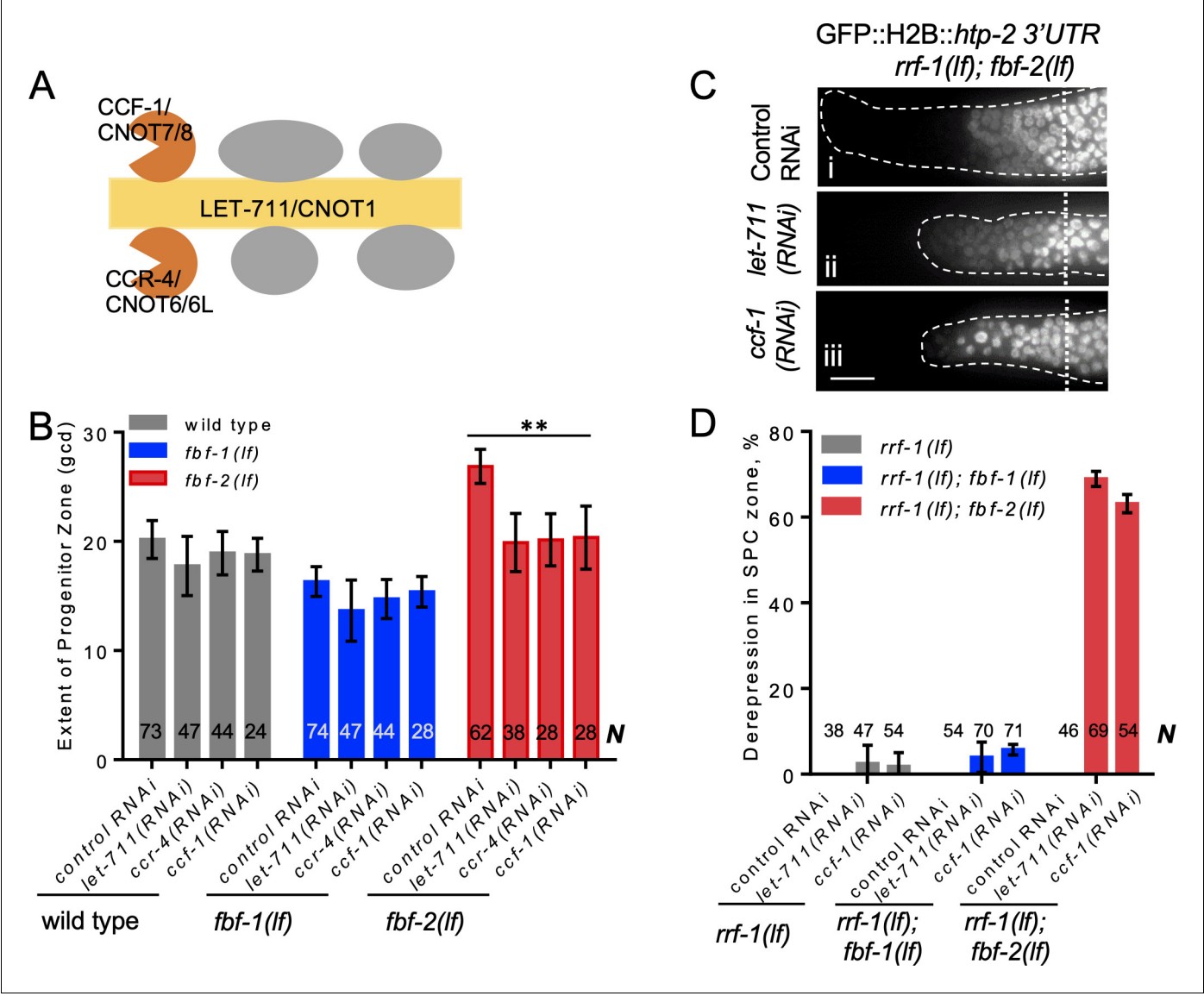

**Figure 3.** CCR4-NOT deadenylase complex promotes FBF-1 function in germline SPCs. (**A**) Schematic of CCR4-NOT deadenylase complex in humans and *C. elegans*; highlighted are the scaffold (yellow) and catalytic (orange) subunits targeted by RNAi in this study. (**B**) The extent of SPC zone after knocking down CCR4-NOT subunits in the wild type, *fbf-1(lf)* and *fbf-2(lf)* genetic backgrounds. Genetic backgrounds and RNAi treatments are indicated on the X-axis and the average size of SPC zone ± SD is plotted on the Y-axis. Differences between CCR4-NOT RNAi and the empty vector RNAi control were evaluated by one-way ANOVA. Asterisks mark the group with significant changes in SPC zone length after CCR4-NOT knockdown, p<0.01. Data were collected from three independent experiments. N, the number of hermaphrodite germlines scored. (**C**) Distal germlines of *rrf-1(lf); fbf-2(lf)* expressing a GFP::Histone H2B fusion under the control of the *htp-2* 3'UTR after the indicated RNAi treatments. Germlines are outlined with dashed lines and vertical dotted lines indicate the beginning of the transition zone. All images were taken with a standard exposure. Scale bar: 10 µm. (**D**) Percentage of germlines showing expression of GFP::H2B fusion extended to the distal end in the indicated genetic backgrounds and knockdown conditions. Plotted values are means ± SD. Data were collected from three independent experiments. *N*, the number of germlines scored. Efficiencies of RNAi treatments were confirmed by sterility (*Figure 3—figure supplement 1B*) or embryonic lethality (*Supplementary file 3*). (B–D) All experiments were performed at 24°C.

The online version of this article includes the following figure supplement(s) for figure 3:

**Figure supplement 1.** CCR4-NOT deadenylase complex promotes FBF-1 function in germline SPCs.

by our knockdown approach. Our findings suggest that CCR4-NOT is required for FBF-1-mediated regulation of germline SPC zone length, but does not significantly contribute to FBF-2 function.

Next, we tested whether CCR4-NOT knockdown disrupts FBF-1-mediated translational repression in SPCs. One FBF target mRNA associated with meiotic entry is *htp-2*, a HORMA domain meiotic protein (*Merritt and Seydoux, 2010*). Translational regulation of a transgenic reporter encoding GFP::Histone H2B fusion under the control of *htp-2* 3'UTR recapitulates FBF-mediated repression in germline SPCs, where GFP::H2B::*htp-2 3'UTR* production is inhibited in the wild type and both *fbf-1* and *fbf-2* single mutant gonads but is strongly derepressed in *fbf-1 fbf-2* double mutant gonads (*Merritt and Seydoux, 2010*). If CCR4-NOT is required for *fbf-1* activity, then *fbf-2(lf)* after CCR4-NOT subunit RNAi should show the same phenotype as *fbf-1(lf) fbf-2(lf)*, or derepression of the reporter. We performed CCR4-NOT RNAi in the *rrf-1(lf)* background to preferentially direct the RNAi effects to the germline and avoid effects on the somatic cells (*Kumsta and Hansen, 2012*; *Sijen et al., 2001*). We observed derepression of the reporter in SPCs of 63–69% germlines of *rrf-1 (lf); fbf-2(lf)* genetic background (*Figure 3C,D*). By contrast, derepression of the reporter was observed only in 3–5% of *rrf-1(lf)* and *rrf-1(lf); fbf-1(lf)* genetic backgrounds (*Figure 3D*; *Figure 3— figure supplement 1A*). These data suggest that the CCR4-NOT deadenylase complex is necessary for FBF-1-mediated repression of target mRNAs in germline SPCs, but is dispensable for FBF-2 regulatory function. *fbf-1 fbf-2* double mutant hermaphrodites are sterile (*Crittenden et al., 2002*). We observed significantly increased sterility upon CCR4-NOT knockdown in *rrf-1(lf); fbf-2(lf)* compared to the *rrf-1(lf)* and *rrf-1(lf); fbf-1(lf)* (*Figure 3—figure supplement 1B*). Like *fbf-1 fbf-2* double mutants, the majority of *rrf-1(lf); fbf-2(lf)* sterile germlines following CCR4-NOT knockdown failed to initiate oogenesis resulting in germline masculinization (data not shown). These observations suggest that CCR4-NOT is required for *fbf-1* activity.

CCR4-NOT knockdown might disrupt FBF-1 regulatory function or FBF-1 protein expression and localization. To distinguish between these possibilities, we determined the abundance of endogenous FBF-1 after *ccf-1(RNAi)* by immunoblot using tubulin as a loading control. We found that FBF-1 protein abundance does not decrease after CCF-1 knockdown compared to the control (*Figure 3— figure supplement 1C,D*). Immunostaining for the endogenous FBF-1 showed that in control germlines FBF-1 localized in foci adjacent to perinuclear P granules (*Figure 3—figure supplement 1E*) as previously reported (*Voronina et al., 2012*). Upon CCF-1 knockdown, FBF-1 foci were still observed next to P granules (*Figure 3—figure supplement 1F*). Therefore, we conclude that CCR4-NOT is not required for FBF-1 expression and localization, and CCR4-NOT knockdown specifically disrupts FBF-1 function.

In summary, we conclude that CCR4-NOT is required for FBF-1, but not FBF-2-mediated regulation of target mRNA and germline SPC zone length. We further predicted that FBF-1 localizes together with CCR4-NOT to the same RNA-protein complex in SPCs.

## FBF-1 colocalizes with CCR4-NOT in germline SPCs

Using co-immunostaining of endogenous FBF-1 or GFP::FBF-1 and 3xFLAG::CCF-1 followed by Pearson's correlation coefficient analysis based on Costes' automatic threshold (*Costes et al., 2004*), we found that both endogenous FBF-1 and GFP::FBF-1 foci colocalize with 3xFLAG::CCF-1 foci in SPC cytoplasm (*Figure 4A,C*; *Figure 4—figure supplement 1A,B*). By contrast, the colocalization between GFP::FBF-2 and 3xFLAG::CCF-1 is significantly less robust (*Figure 4B,C*). As an alternative metric of colocalization, we used proximity ligation assay (PLA) that can detect protein-protein interactions in situ at the distances < 40 nm (*Fredriksson et al., 2002*). PLA was performed in *3xflag::ccf-1; gfp::fbf-1*, *3xflag::ccf-1; gfp::fbf-2*, and *3xflag::ccf-1; gfp* animals using the same antibodies and conditions for all three protein pairs. We observed significantly more dense PLA signals in *3xflag::ccf-1; gfp::fbf-1* than in the control (*Figure 4D*; p<0.0001, *Table 2*). By contrast, PLA foci density in mitotic germ cells of *3xflag::ccf-1; gfp::fbf-2* was not different from the control (*Figure 4D*; *Table 2*), although the expression of GFP::FBFs or GFP alone in mitotic germ cells appeared similar (*Figure 4—figure supplement 1C*). Together, these data suggest that FBF-1, but not FBF-2, colocalizes with CCR4-NOT in SPCs, in agreement with the dependence of FBF-1 function on CCR4-NOT.

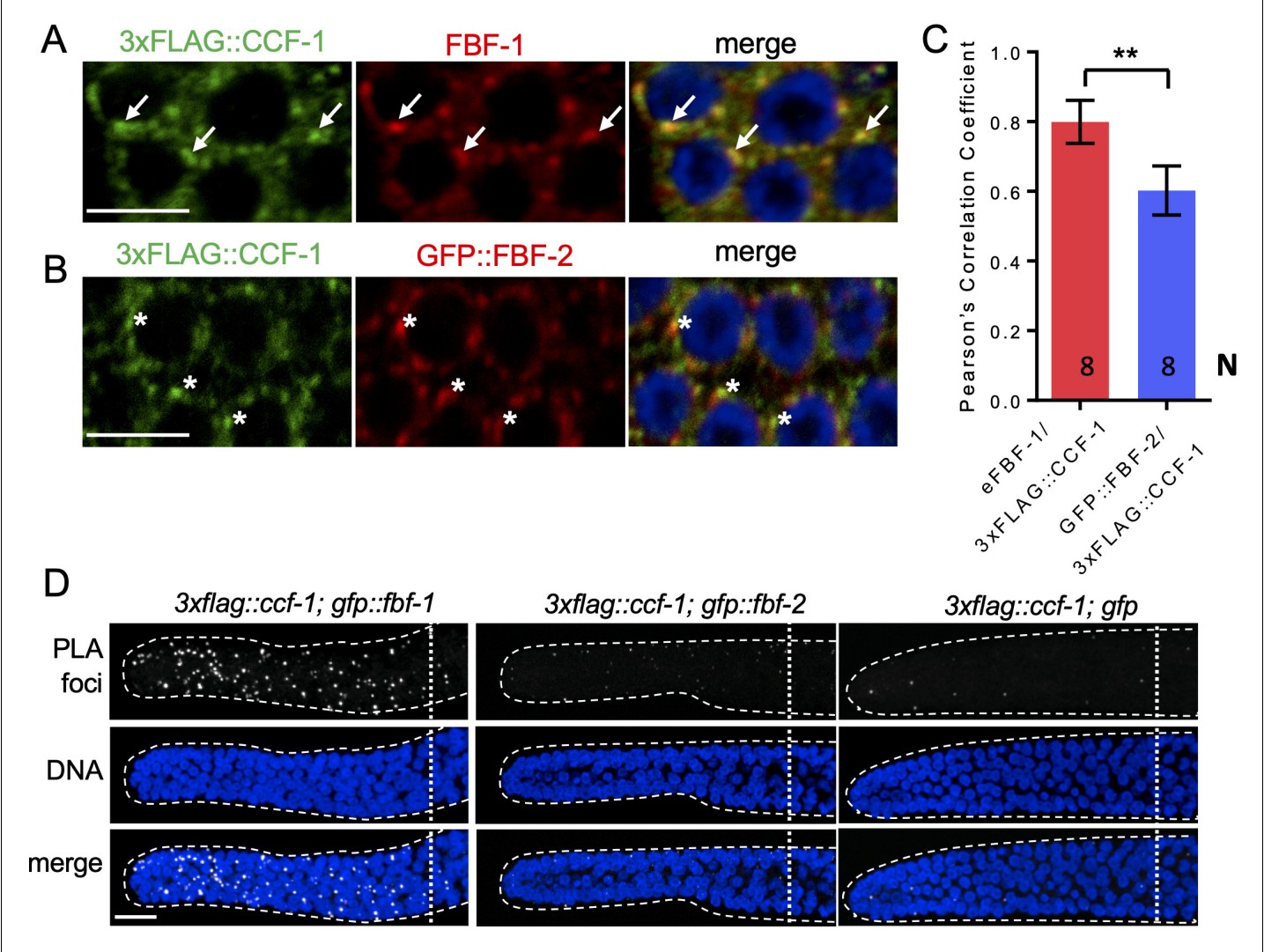

**Figure 4.** FBF-1 colocalizes with CCR4-NOT complex in germline SPCs. (A–B) Confocal images of SPCs co-immunostained for endogenous FBF-1 (A) or GFP-tagged FBF-2 (B, red) and 3xFLAG-tagged CCF-1 (green). DNA staining is in blue (DAPI). Arrows indicate complete overlap of FBF-1 and CCF-1 granules. Asterisks denote FBF-2 granules localizing close but not overlapping with CCF-1 granules. Scale bars in A and B: 5 μm. (C) Pearson's correlation analysis quantifying the colocalization between FBF and CCF-1 granules in co-stained germline images. Plotted values are means ± SD. *N*, the number of analyzed germline images (single confocal sections through the middle of germline SPC nuclei including 5–8 germ cells). Asterisks mark statistically significant difference by Student's t-test, p<0.01. (D) Confocal images of the distal germline SPC zones with PLA foci (grayscale) and DNA staining (blue). Germlines are outlined with dashed lines and vertical dotted lines indicate the beginning of the transition zone. Genotypes are indicated on top of each image group. Scale bar: 10 μm. (A–D) All experiments were performed at 24˚C.

The online version of this article includes the following figure supplement(s) for figure 4:

**Figure supplement 1.** FBF-1 colocalizes with CCR4-NOT complex in germline SPCs.

## FBF-1 and FBF-2 differentially impact target mRNAs polyadenylation

Since a knockdown of CCR4-NOT deadenylase produces distinct effects on FBF-1- and FBF-2-mediated target repression, we hypothesized that FBFs differentially affect deadenylation of target mRNAs. To test this hypothesis, we compared the length of the poly(A) tail of two FBF target mRNAs among the wild type, *fbf-1(lf)*, and *fbf-2(lf)* genetic backgrounds by Poly(A) tail (PAT)-PCR. We selected the targets associated with both cell cycle (*cyb-2.1; Kershner and Kimble, 2010; Porter et al., 2019; Prasad et al., 2016*) and meiotic entry (*htp-1; Merritt and Seydoux, 2010*) and used an mRNA not associated with FBFs (*unc-54*) as a control. RNA samples were extracted from animals of *glp-1 (gain-of-function, gf)* mutant background, which produces germlines with a large

**Table 2.** Proximity ligation assay detects association of FBF-1 with CCR4-NOT complex component CCF-1.

PLA foci density was determined in maximal intensity projections of confocal image stacks encompassing germline SPC zones of the indicated strains. Reported values are mean ± SD derived from three independent biological replicates (*3xflag::ccf-1; gfp::fbf-1* and *3xflag::ccf-1; gfp::fbf-2*) or a single replicate (*3xflag::ccf-1; gfp*), all reared at 24°C. Differences in PLA density between *3xflag::ccf-1; gfp::fbf-1* or *3xflag::ccf-1; gfp::fbf-2* and the control *3xflag::ccf-1; gfp* were analyzed by one-way ANOVA with Dunnett's post-test. N, number of germline images analyzed.

| Genotype | PLA density in SPC zone (/um2) x $10^{-2}$ | P value,vs. control | N |
|---|---|---|---|
| *3xflag::ccf-1; gfp::fbf-1* | 5.2 ± 2.4 | <0.0001 | 32 |
| *3xflag::ccf-1; gfp::fbf-2* | 1.1 ± 0.8 | ns | 27 |
| *3xflag::ccf-1; gfp* | 0.6 ± 0.2 | n/a | 12 |

number of mitotic cells at the restrictive temperature (*Pepper et al., 2003a*; *Figure 5—figure supplement 1A*), thus allowing us to focus on the mRNAs in the mitotic cell population.

PAT-PCR assays using RNA samples extracted from *fbf-1(lf); glp-1(gf)*, *glp-1(gf)*, and *fbf-2(lf); glp-1(gf)* revealed that the poly(A) tail distributions of *cyb-2.1* and *htp-1* mRNAs in *fbf-2(lf)* were both shifted to shorter lengths compared to the wild type background (*Figure 5A,B,D,E*). Conversely, in *fbf-1(lf)* background, the poly(A) tail profiles of *cyb-2.*1 and *htp-1* mRNAs were shifted to longer lengths (*Figure 5A,B,D,E*). By contrast, the poly(A) tail of the control *unc-54* mRNA did not decrease in *fbf-2(lf)* background or increase in *fbf-1(lf)* compared to the wild type (*Figure 5C,F*). We conclude that FBF-1 promotes deadenylation of FBF target mRNAs and FBF-2 protects the targets from deadenylation. This is consistent with weaker effects of CCR4-NOT knockdown on FBF target regulation in the genetic backgrounds where FBF-2 is present (wild type and *fbf-1(lf)*, *Figure 3*).

Cytoplasmic deadenylation of mRNA frequently leads to its decay (*Mugridge et al., 2018*). To test whether differential polyadenylation of FBF targets in *fbf-1(lf)* and *fbf-2(lf)* resulted in changes in their steady-state amounts relative to the wild type, we compared the mRNA abundance of several FBF targets among *fbf-1(lf); glp-1(gf)*, *glp-1(gf)*, and *fbf-2(lf); glp-1(gf)* genetic backgrounds by qPCR (*Figure 5—figure supplement 1B*). We determined steady-state levels of both meiotic entry-associated transcripts *him-3*, *htp-1*, and *htp-2* (*Merritt and Seydoux, 2010*) and cell cycle regulators *cyb-1*, *cyb-2.1*, *cyb-2.2* and *cyb-3* (*Kershner and Kimble, 2010*; *Porter et al., 2019*; *Prasad et al., 2016*), and used *unc-54* as a control. All transcript levels were normalized to a housekeeping gene actin (*act-1*). The steady-state abundance of FBF targets in *fbf-2(lf)* decreased relative to the wild type and the decrease was significant for all transcripts except *him-3* (*Figure 5—figure supplement 1B*). The levels of most FBF targets in the *fbf-1(lf)* were not significantly different from those in the wild-type background, except for *cyb-2.1* that accumulated to significantly lower levels (*Figure 5—figure supplement 1B*). By contrast, the abundance of *unc-54* mRNA did not decrease in either genetic background. We conclude that shorter poly(A) tails of FBF targets in *fbf-2(lf)* are associated with lower mRNA accumulation for many targets.

## Three variable regions outside of FBF-2 RNA-binding domain are necessary to prevent cooperation with CCR4-NOT

Our findings suggest that FBF-1-mediated SPC maintenance depends on the CCR4-NOT deadenylase complex, while FBF-2 can function independent of CCR4-NOT. Since FBF proteins are very similar in primary sequence except for the four variable regions (VRs, *Figure 6A*), we next investigated whether the VRs were necessary for FBF-2-specific maintenance of germline SPCs and prevented FBF-2 dependence on CCR4-NOT. We previously found that mutations/deletions of the VRs outside of FBF-2 RNA-binding domain (VR1, 2 and 4, *Figure 6A*) produced GFP::FBF-2(vrm) protein with a disrupted localization and compromised function (*Wang et al., 2016*). We hypothesized that these three VRs might contribute to FBF-2-specific effects on the extent of SPC zone as well as prevent FBF-2 from cooperating with CCR4-NOT.

We first tested whether the three VRs are required for FBF-2-specific SPC zone length. To test this hypothesis, the extent of SPC zone was determined after crossing the GFP::FBF-2(vrm)

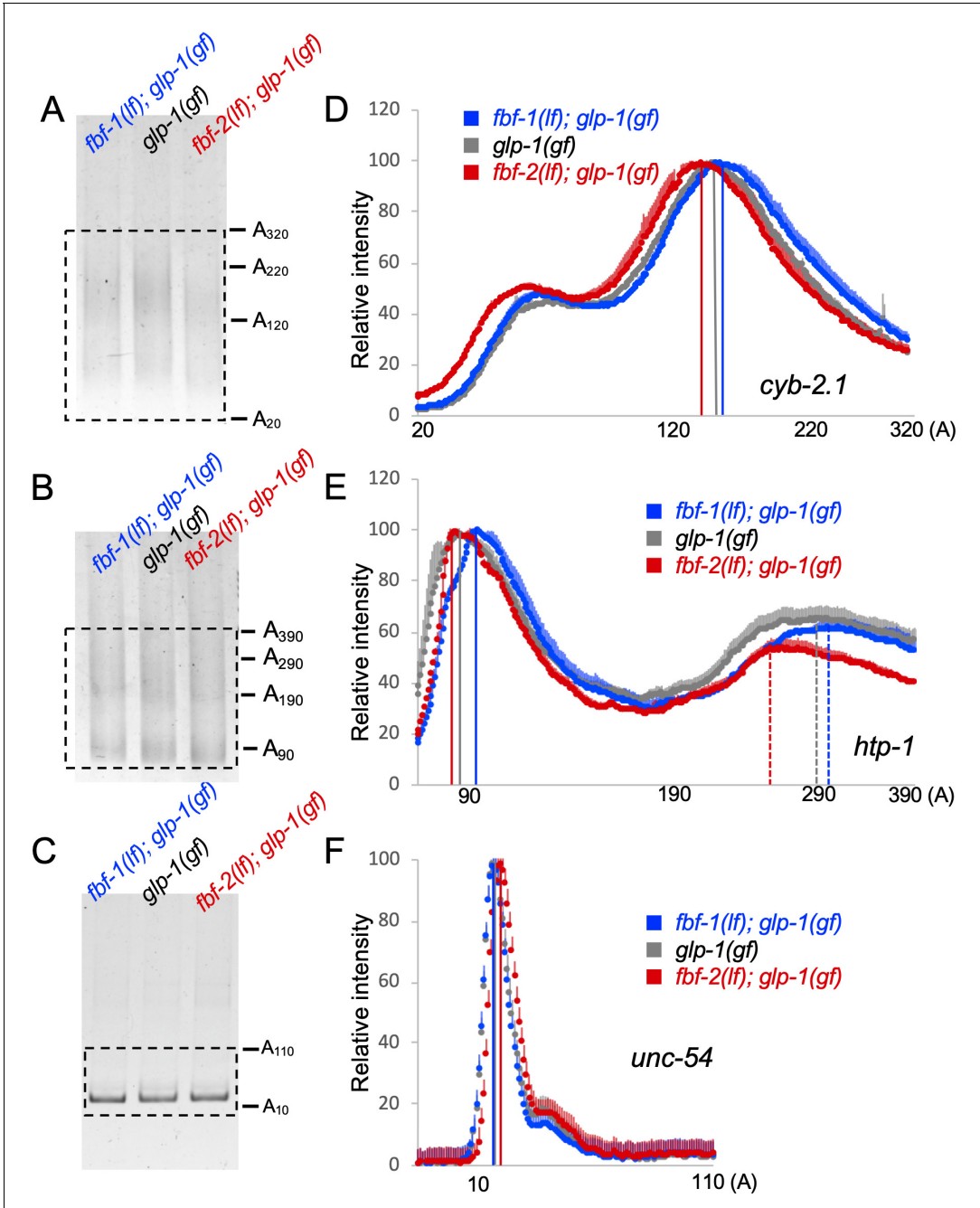

**Figure 5.** Antagonistic effects of FBF-1 and FBF-2 on poly(A) tail of target mRNAs. (**A–C**) Representative PAT-PCR amplification of the poly(A) tail of *cyb-2.1* (**A**), *htp-1* (**B**) and control myosin heavy chain *unc-54* (**C**) in *fbf-1(lf); glp-1(gf), glp-1(gf)*, and *fbf-2(lf); glp-1(gf)* genetic backgrounds at 25°C. The estimated lengths of poly(A) tails based on the PCR fragment sizes are indicated on the right. The areas boxed by dotted lines were quantified by densitometry in ImageJ. (**D–F**) Densitometric quantification of PAT-PCR amplification products (boxed in A-C). Y-axis, mean relative intensity represents the average intensities of normalized PAT-PCR reactions from three independent biological replicates. X-axis, estimated sizes of poly(A) tails. Values are means + SD. Vertical lines in (**D–F**) mark the peaks of PAT-PCR intensity profiles for each mRNA in each genetic background, dashed lines mark secondary peaks for *htp-1*. (**A–F**) Nematodes for all replicates were grown at 25°C.

The online version of this article includes the following figure supplement(s) for figure 5:

**Figure supplement 1.** RNA analysis from predominantly mitotic germlines in *glp-1(gf)* genetic background.

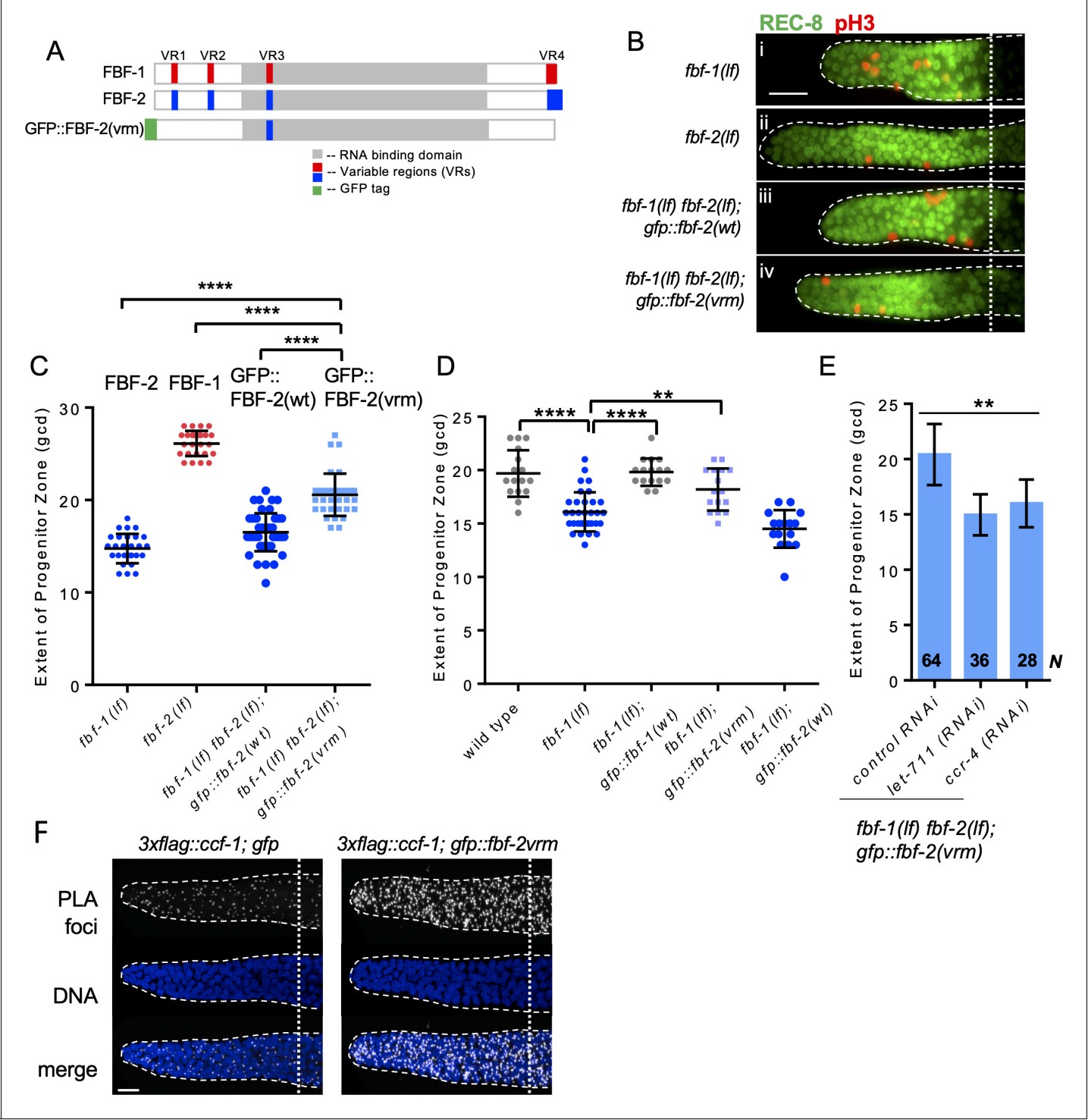

**Figure 6.** Three variable regions of FBF-2 prevent its cooperation with CCR4-NOT. (**A**) Schematics of FBF-1, FBF-2 and GFP::FBF-2(vrm) mutant transgene (*Wang et al., 2016*). Red and blue boxes indicate variable regions distinguishing FBF-1 and FBF-2 respectively, grey box indicates the RNA-binding domain, and green box indicates GFP tag. (**B**) Distal germlines of the indicated genetic backgrounds stained with anti-REC-8 (green) and anti-pH3 (red). Germlines are outlined with the dashed lines, and the vertical dotted line marks the beginning of transition zone. Scale bar: 10 μm. (**C**) The extent of SPC zone in the indicated genetic backgrounds (on the X-axis). FBF protein(s) present in each genetic background are noted above each data set. Plotted values are individual data points and means ± SD. Differences in SPC zone length between *fbf-1(lf) fbf-2(lf); gfp::fbf-2(vrm)* and the other strains were evaluated by one-way ANOVA test with Dunnett's post-test; asterisks mark statistically significant differences (p<0.0001). Data were collected from three independent experiments and 24–28 germlines were scored for each genotype. (**D**) The extent of SPC zone was measured after

*Figure 6 continued on next page*

*Figure 6 continued*

crossing the GFP::FBF-2(vrm), GFP::FBF-1(wt) and GFP::FBF-2(wt) transgenes into *fbf-1(lf)* genetic background. As controls, SPC zone length was also measured in *fbf-1(lf)* and the wild type. Plotted values are individual data points and means ± SD. Data were collected from two independent experiments and 8–17 germlines were scored for each genotype per replicate. Differences in SPC zone length between *fbf-1(lf)* and all other strains were evaluated by one-way ANOVA test with Dunnett's post-test; asterisks mark statistically significant differences (****, p<0.0001; **p<0.01). (E) The extent of SPC zone after knocking down CCR4-NOT subunits in the *fbf-1(lf) fbf-2(lf); gfp::fbf-2(vrm)* genetic background. RNAi treatments are indicated on the X-axis and average length of SPC zone ± SD on the Y-axis. Differences in SPC zone length between CCR4-NOT knockdowns and control were evaluated by one-way ANOVA (asterisks, p<0.01). Data were collected from three independent experiments. N, the number of independent germlines scored. (F) Confocal images of the distal germline SPC zones with PLA foci (grayscale) and DNA staining (blue). Germlines are outlined with dashed lines and vertical dotted lines indicate the beginning of the transition zone. Genotypes are indicated on top of each image group. Scale bar: 10 μm. (B–D): All experiments were performed at 24°C.

The online version of this article includes the following figure supplement(s) for figure 6:

**Figure supplement 1.** Variable regions 1, 2 and 4 of FBF-2 are required to rescue FBF-2-specific function in germline SPCs.

transgene into *fbf* double mutant background. We found that the SPC zone maintained by GFP::FBF-2(vrm) (*Figure 6Biv*) is significantly longer than that maintained by GFP::FBF-2(wt) (*Figure 6Biii*) and the endogenous FBF-2 (p<0.01; *Figure 6Bi*, C) suggesting that the GFP::FBF-2(vrm) effect on SPC zone length is distinct from that of FBF-2. Western blot analysis indicated that expression of GFP::FBF-2(vrm) is comparable to that of GFP::FBF-2(wt), so their distinct effects on SPC zone length are likely due to functional differences (*Figure 6—figure supplement 1A*). To test whether GFP::FBF-2(vrm) can rescue either of *fbf* single mutants, we determined the extent of SPC zone after crossing GFP::FBF-2(vrm) into *fbf-1(lf)* and *fbf-2(lf)* genetic backgrounds. As controls, the lengths of SPC zones were also measured after crossing the wild type GFP::FBF-2(wt) and GFP::FBF-1(wt) transgenes into each *fbf* single mutant. As expected, the SPC zone length of *fbf-2(lf); gfp::fbf-2(wt)* is significantly shorter than *fbf-2(lf)* (p<0.01) while the SPC zone of *fbf-2(lf); gfp::fbf-1(wt)* is similar to *fbf-2(lf)* (*Figure 6—figure supplement 1B*), suggesting that GFP::FBF-2(wt), but not GFP::FBF-1(wt), rescues *fbf-2(lf)*. Likewise, GFP::FBF-1(wt), but not GFP::FBF-2(wt), rescues *fbf-1(lf)* (p<0.01, *Figure 6D*). Interestingly, we found that the extent of SPC zone of *fbf-2(lf); gfp::fbf-2(vrm)* is similar to that of *fbf-2(lf)* (*Figure 6—figure supplement 1B*), suggesting that GFP::FBF-2(vrm) does not rescue *fbf-2(lf)*. By contrast, the SPC zone of *fbf-1(lf); gfp::fbf-2(vrm)* is significantly longer than that of *fbf-1(lf)* (p<0.01, *Figure 6D*) and there is no significant difference in the SPC zone length between *fbf-1(lf); gfp::fbf-2(vrm)* and the wild type, suggesting that the GFP::FBF-2(vrm) rescues *fbf-1(lf)*. We conclude that the three VRs outside of FBF-2 RNA-binding domain (VR1, 2, and 4) are important for FBF-2-specific effect on the extent of germline SPC zone and mutation or deletion of these VRs resulted in a mutant protein FBF-2(vrm) that functions similar to FBF-1.

Since FBF-1 function requires the CCR4-NOT complex and FBF-2(vrm) appears similar to FBF-1, we hypothesized that CCR4-NOT is required for FBF-2(vrm) function. To test this hypothesis, we measured SPC zone length after knockdown of CCR4-NOT subunits in *fbf-1(lf) fbf-2(lf); gfp::fbf-2(vrm)* animals by RNAi. We found that SPC zone of *fbf-1(lf) fbf-2(lf); gfp::fbf-2(vrm)* after RNAi of CCR4-NOT subunits becomes significantly shorter than the control (p<0.01, *Figure 6E*), suggesting that GFP::FBF-2(vrm) function requires CCR4-NOT. Furthermore, if GFP::FBF-2(vrm) cooperates with CCR4-NOT, we expect that it might associate with CCF-1 by proximity ligation assay. Indeed, PLA foci density in the mitotic cells of *3xflag::ccf-1; gfp::fbf-2(vrm)* was significantly greater than in the control (*Figure 6F*; *Table 3*; p<0.0001). We conclude that the VRs outside of FBF-2 RNA-binding domain are required for FBF-2-specific effect on the extent of SPC zone and to prevent FBF-2 from cooperating with CCR4-NOT.

## The variable region 4 (VR4) of FBF-2 is sufficient to prevent cooperation with CCR4-NOT

To test whether one of the three VRs outside of FBF-2 RNA-binding domain (VR1, 2, and 4) is sufficient to support FBF-2-specific effects on the length of SPC zone, we established a transgenic FBF-1 chimera with VR4 swapped from FBF-2 (GFP::FBF-1(FBF-2vr4); *Figure 7A*) and crossed it into *fbf* double mutant. Since VR3 residing in FBF-2 RNA-binding domain was not sufficient for FBF-2-specific function, *gfp::fbf-1(fbf-2vr3)* (with VR3 swapped from FBF-2; *Figure 7A*) chimeric transgene was made for comparison. SPC zone length assessment showed that the SPC zone maintained by GFP::

**Table 3.** Proximity ligation assay detects association of FBF-2(vrm) with CCR4-NOT complex component CCF-1.

PLA foci density was determined in maximal intensity projections of confocal image stacks of germline SPC zones of the indicated strains. Reported values are mean ± SD derived from three independent biological replicates reared at 24°C. PLA densities were compared by Student's t-test. *N*, number of germline images analyzed.

| Genotype | PLA density in SPC zone (/um2) x $10^{-1}$ | p value, vs. control | N |
|---|---|---|---|
| *3xflag::ccf-1; gfp::fbf-2vrm* | 2.3 ± 0.7 | <0.0001 | 58 |
| *3xflag::ccf-1; gfp* | 1.1 ± 0.5 | n/a | 48 |

FBF-1(FBF-2vr4) (*Figure 7Biii*) is significantly shorter than that maintained by GFP::FBF-1(wt) (*Figure 7Bv*) and endogenous FBF-1 (p<0.0001; *Figure 7Bii*, C). By contrast, the SPC zone maintained by GFP::FBF-1(FBF-2vr3) (*Figure 7Biv*) is similar to that maintained by the GFP::FBF-1(wt) (*Figure 7Biv*, C). This finding suggested that GFP::FBF-1(FBF-2vr4) might function similarly to FBF-2. Western blot analysis revealed that the protein expression levels of GFP::FBF-1, GFP::FBF-1(FBF-2vr3), and GFP::FBF-1(FBF-2vr4) were comparable, so the differential effects on SPC zone length are likely due to functional differences (*Figure 7—figure supplement 1A*). To test whether GFP::FBF-1(FBF-2vr4) rescues FBF-1- or FBF-2-specific function, we measured the extent of SPC zones after crossing GFP::FBF-1(FBF-2vr4) into *fbf-1(lf)* and *fbf-2(lf)* genetic backgrounds. For comparison, GFP::FBF-1(FBF-2vr3) was also crossed into each *fbf* single mutant. We found that the SPC zone of *fbf-1 (lf); gfp::fbf-1(fbf-2vr4)* is similar to that of *fbf-1(lf)* (*Figure 7—figure supplement 1B*), suggesting that GFP::FBF-1(FBF-2vr4) does not rescue *fbf-1(lf)*. Interestingly, SPC zone of *fbf-2(lf); gfp::fbf-1(fbf-2vr4)* is significantly shorter than that of *fbf-2(lf)* (p<0.01, *Figure 7—figure supplement 1C*), suggesting that GFP::FBF-1(FBF-2vr4) rescues *fbf-2(lf)*. By contrast, GFP::FBF-1(FBF-2vr3) rescues *fbf-1 (lf)*, but not *fbf-2(lf)* (*Figure 7—figure supplement 1B,C*). We conclude that the presence of VR4 from FBF-2 in a chimeric GFP::FBF-1(FBF-2vr4) protein is sufficient to impart FBF-2-specific effect on the extent of SPC zone.

To test whether VR4 is sufficient to inhibit cooperation of GFP::FBF-1(FBF-2vr4) with CCR4-NOT, we measured the length of SPC zone after knockdown of CCR4-NOT subunits in *fbf-1(lf) fbf-2(lf); gfp::fbf-1(fbf-2vr4)* animals by RNAi. As a control, CCR4-NOT knockdown was also performed on *fbf-1(lf) fbf-2(lf); gfp::fbf-1(fbf-2vr3)*. We found that the SPC zone of *fbf-1(lf) fbf-2(lf); gfp::fbf-1(fbf-2vr4)* after RNAi of CCR4-NOT subunits is similar to the control (*Figure 7D*), suggesting that GFP::FBF-1(FBF-2vr4) function in SPCs does not rely on CCR4-NOT. By contrast, the SPC zone of *fbf-1(lf) fbf-2(lf); gfp::fbf-1(fbf-2vr3)* is significantly shortened after RNAi of CCR4-NOT subunits compared to the control (p<0.01, *Figure 7D*), indicating that GFP::FBF-1(FBF-2vr3) maintains dependence on CCR4-NOT. We conclude that FBF-2 VR4 in a chimeric GFP::FBF-1(FBF-2vr4) protein is sufficient to support FBF-2-specific effect on the extent of germline SPC zone and to prevent the chimera's cooperation with CCR4-NOT.

## Discussion

Our results support three main conclusions that advance our understanding of how PUF family FBF proteins modulate rates of cell cycle progression and meiotic entry in *C. elegans* germline stem and progenitor cells. First, FBF proteins simultaneously adjust SPC cell cycle and meiotic entry rates through regulation of FBF target mRNAs affecting each process. Second, FBF-mediated repression of cyclin B affects SPC cell division rate. Third, distinct effects of FBF homologs on their target mRNAs and SPC development are mediated by differential cooperation of FBFs with deadenylation machinery. In turn, activation of deadenylation machinery by FBFs depends on the protein sequences outside of the conserved PUF RNA-binding domain. Collectively, our results support a model where two paralogous FBF proteins achieve complementary effects on SPC cell division and meiotic entry through distinct regulatory mechanisms (*Figure 8*).

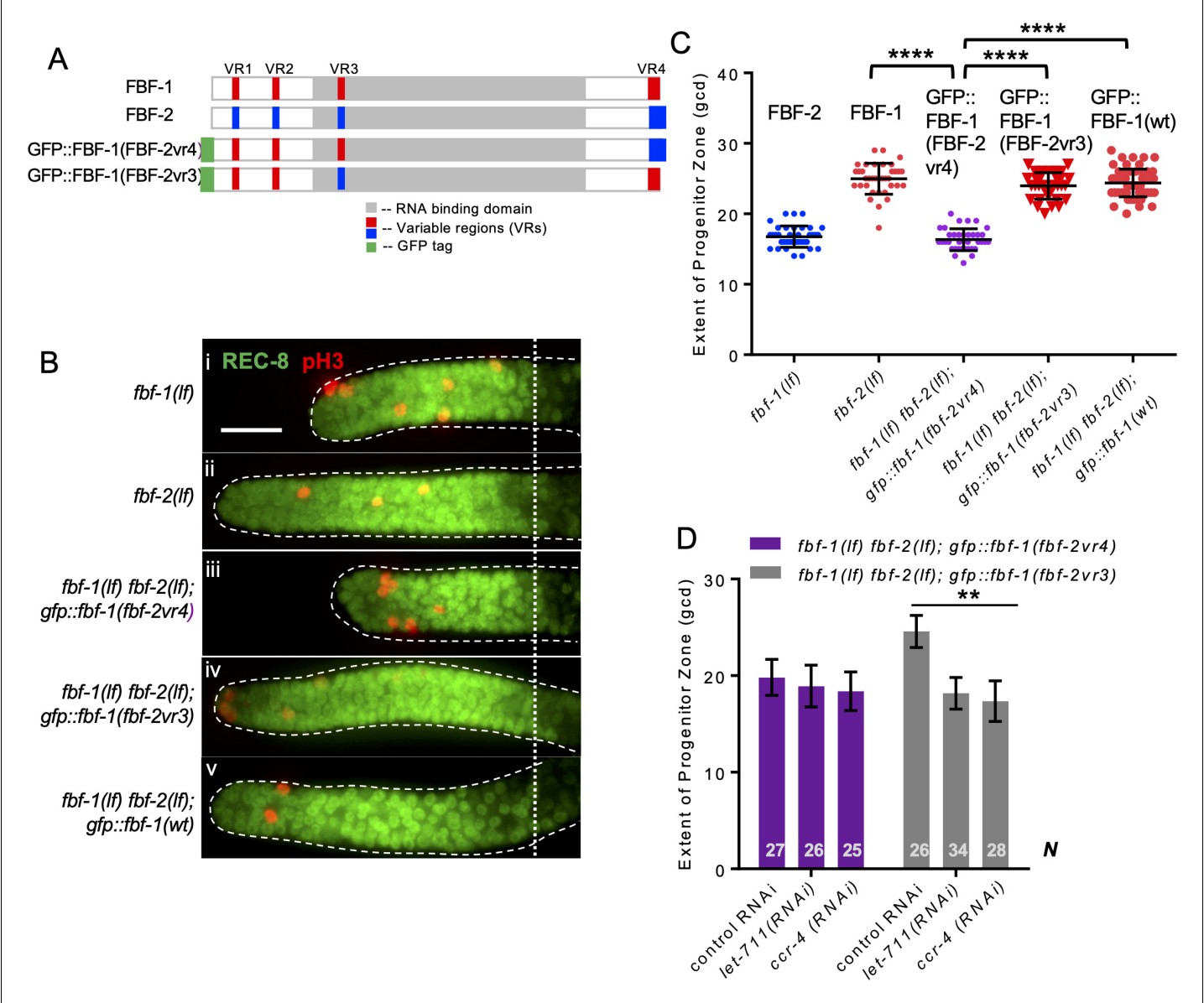

**Figure 7.** Variable region 4 (VR4) from FBF-2 is sufficient to prevent FBF-1 chimera from cooperation with CCR4-NOT. (**A**) Schematics of FBF-1, FBF-2, transgenic GFP::FBF-1(FBF-2vr4) chimera (with VR4 swapped from FBF-2), and transgenic GFP::FBF-1(FBF-2vr3) chimera (with VR3 swapped from FBF-2). Red and blue boxes, variable regions distinguishing FBF-1 and FBF-2 respectively; grey box, RNA-binding domain; green box, GFP tag. (**B**) Distal germlines dissected from the indicated genetic backgrounds stained with anti-REC-8 (green) and anti-pH3 (red). Germlines are outlined with the dashed lines and the vertical dotted line marks the beginning of the transition zone. Scale bar: 10 μm. (**C**) The extent of SPC zone in the indicated genetic backgrounds (on the X-axis). FBF protein present in each genetic background is noted above each data set. Plotted values are individual data points and means ± SD. Differences in SPC zone length between *fbf-1(lf) fbf-2(lf); gfp::fbf-1(fbf-2vr4)* and the other strains were evaluated by one-way ANOVA test with Dunnett's post-test; asterisks mark statistically significant differences (p<0.0001). Data were collected from three independent experiments and 31–60 germlines were scored for each genotype. (**D**) SPC zone length after knocking down CCR4-NOT subunits in the *fbf-1(lf) fbf-2(lf); gfp::fbf-1(fbf-2vr4)* and *fbf-1(lf) fbf-2(lf); gfp::fbf-1(fbf-2vr3)* genetic backgrounds (as indicated on the X-axis). Plotted values are means ± SD. Asterisks mark the group with significant changes in SPC zone length after CCR4-NOT knockdown vs control RNAi by one-way ANOVA (p<0.01). Data were collected from two independent experiments. N, the number of hermaphrodite germlines scored. (**B–D**) All experiments were performed at 24°C. The online version of this article includes the following figure supplement(s) for figure 7:

**Figure supplement 1.** Variable region 4 of FBF-2 allows chimeric FBF-1vr4 to rescue *fbf-2(lf)*.

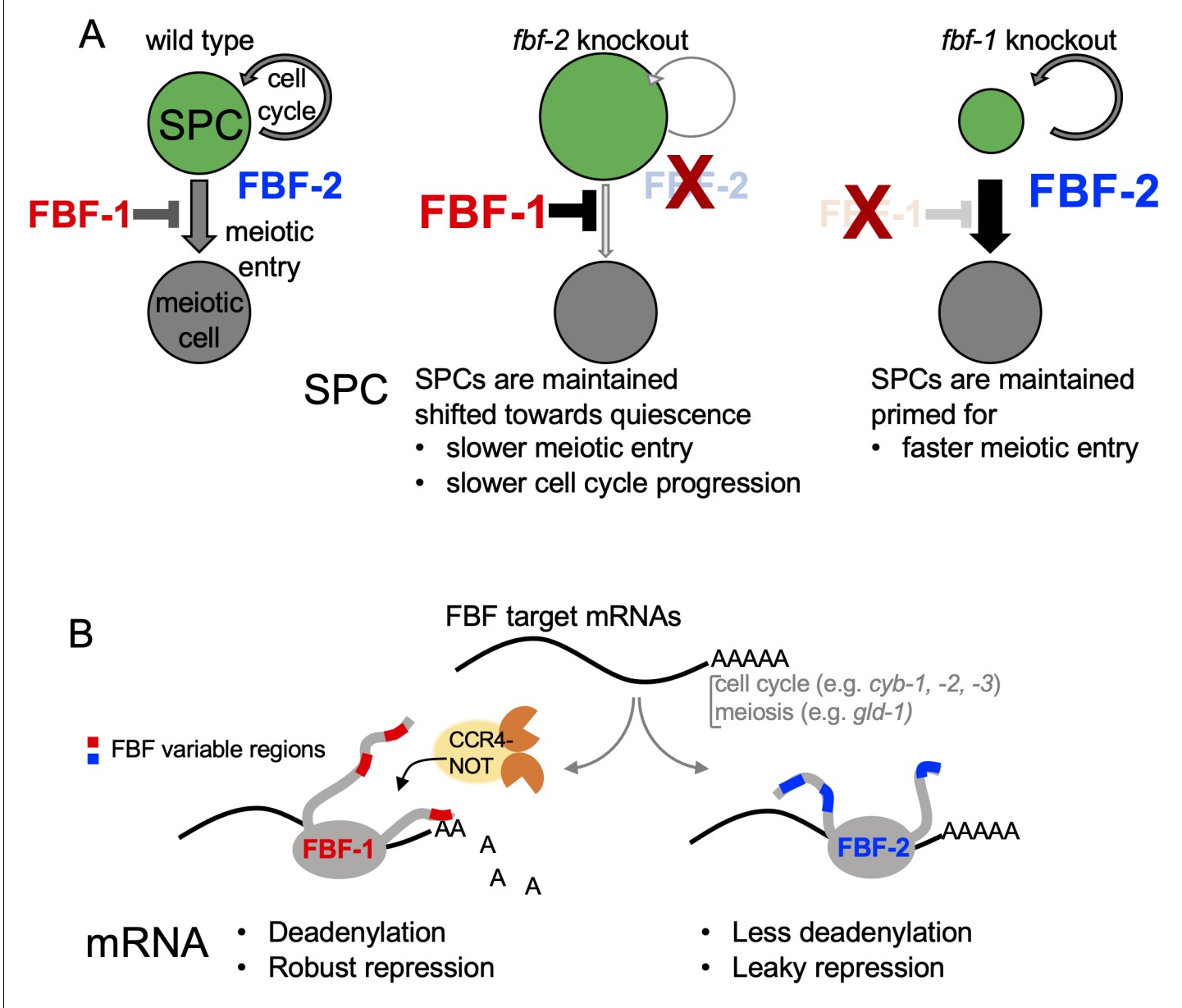

**Figure 8.** A model of antagonistic effects of FBF-1 and FBF-2 on germline SPC dynamics and target mRNAs. (**A**) FBFs regulate the rate of stem and progenitor (green) cell division and the rate of entry into meiosis (grey). Both FBFs negatively regulate each other's expression (**Lamont et al., 2004**, not pictured). In *fbf-2(lf)*, FBF-1 is overexpressed, and both cell division and meiotic entry rates are slow. This results in an increased total number of SPCs in the adult (larger green circle). In *fbf-1(lf)*, FBF-2 is overexpressed and meiotic entry rate is enhanced, reducing overall numbers of SPCs (smaller green circle). In the double *fbf-1(lf) fbf-2(lf)* mutant (not pictured), adult SPCs are lost to differentiation. (**B**) FBF-1 and FBF-2 bind the same target mRNAs, and each can promote SPC maintenance. FBF-1 cooperates with CCR4-NOT deadenylase and robustly represses target mRNAs (**Crittenden et al., 2002**; **Brenner and Schedl, 2016**) to restrict the rate of germline stem cell meiotic entry. FBF-2 protects mRNAs from deadenylation; FBF-2-dependent regulation is associated with leaky repression of protein synthesis and sustains the wild-type rates of SPC cell division and meiotic entry. Differential cooperation of FBFs with CCR4-NOT is determined by their variable regions outside of the RNA-binding domain.

## FBFs affect the rates of both stem cell mitotic divisions and meiotic entry

Here, we provide evidence that loss-of-function mutation of *fbf* paralogs change the rates of both cell cycle and meiotic entry in *C. elegans* germline SPC. We find that slow proliferation of SPCs in *fbf-2(lf)* is associated with a slower rate of progenitor meiotic entry (differentiation), while the progenitors of *fbf-1(lf)* mutant have a faster rate of meiotic entry (**Figure 1D–F**). We propose that cell

cycle and meiotic entry are coordinately affected by FBF-mediated control of target mRNAs relevant to each process (*Figure 8*). Antagonistic post-transcriptional regulation by FBF-1 and FBF-2 promotes the wild-type coordination of cell cycle progression and meiotic entry. Loss of either *fbf* changes the stem cell system dynamics, which in turn results in changes of SPC zone size in the individual *fbf* mutants compared to the wild type (*Figure 8A*). Nevertheless, each individual FBF is able to maintain germline stem cells, in a state that is either closer to quiescent (FBF-1), or primed for differentiation (FBF-2). Although this model is likely oversimplified, it provides a framework for future exploration of factors controlling SPC dynamics.

Previous research identified SYGL-1 and LST-1 as key transcriptional targets of GLP-1/NOTCH that promote germline stem cell fate (*Kershner et al., 2014*). SYGL-1 and LST-1 are cofactors of both FBFs that promote FBF-dependent regulation (*Haupt et al., 2019*; *Shin et al., 2017*). The abundance of SYGL-1 and LST-1 impacts the number of uncommitted stem cells and thus the overall size of SPC zone (*Haupt et al., 2019*; *Shin et al., 2017*). Remarkably, both SYGL-1 and LST-1 are also FBF targets (*Kershner and Kimble, 2010*; *Kershner et al., 2014*; *Prasad et al., 2016*), and it would be interesting to investigate whether changes in SPC zone length in *fbf* mutants are associated with altered levels or pattern of expression of SYGL-1 and/or LST-1.

Slow meiotic entry rate in *fbf-2(lf)* likely results from enhanced translational repression of FBF targets that regulate differentiation; indeed, slower accumulation of the FBF target GLD-1 has been documented in this genetic background (*Brenner and Schedl, 2016*). In a similar fashion, mutations of FBF targets *gld-2* and *gld-3* lead to a decrease in meiotic entry rate and to accumulation of excessive numbers of SPCs (*Eckmann et al., 2004*; *Fox and Schedl, 2015*). Conversely, higher meiotic entry rate of *fbf-1(lf)* SPCs might be explained by partial derepression of FBF targets. FBF-2 represses FBF target transcripts in *fbf-1(lf)* background, while sequestering them in large cytoplasmic aggregates (*Voronina et al., 2012*). However, this repression is less effective than that exerted by FBF-1 since partial derepression of GLD-1 has been previously observed in *fbf-1(lf)* (*Brenner and Schedl, 2016*; *Crittenden et al., 2002*).

We find that FBF-2 promotes SPC cell division by facilitating progression through the G2-phase of cell cycle (*Figure 1D*). Thus, SPCs of the *fbf-2(lf)* mutant are characterized by longer median G2-phase length. By contrast, the G2-phase of *fbf-1(lf)* SPCs is the same as in the wild type, even though this genetic background shows an increase in the mitotic index (*Figure 1D*, *Figure 1—figure supplement 1B*). There are several possible explanations for a higher mitotic index in *fbf-1(lf)* background. One is slow progression through the M-phase of the cell cycle. We tested for activation of mitotic checkpoints in *fbf-1(lf)* genetic background, but found no difference with the wild type in the prevalence of inactivated NCC-1/CDK-1 (pTyr15) (data not shown). Another possibility for the increase in mitotic index is the difference in the proportion of cycling-competent versus non-cycling cells in *fbf-1(lf)*. The proximal SPC zone contains non-cycling cells in meiotic S-phase, estimated to comprise 30–40% of total SPC zone cells in the wild-type germlines (*Crittenden et al., 2006*; *Jaramillo-Lambert et al., 2007*; *Fox et al., 2011*). Faster meiotic entry rate of *fbf-1(lf)* SPCs might be associated with faster progenitor transit through the meiotic S-phase. As a result, the number of non-cycling premeiotic cells (and consequently the total number of SPC zone cells) would be lower, leading to an inflated SPC mitotic index. We could not address whether *fbf-1(lf)* germlines have a lower number of progenitors in meiotic S-phase since there are no specific molecular markers for this developmental stage. Finally, we find that disruption of FBF-mediated regulation of a single B-type cyclin in slowly proliferating and slowly differentiating *fbf-2(lf)* SPCs is sufficient to disturb stem cell homeostasis, to promote faster cell cycle, and to result in excessive SPC accumulation (*Figure 2*). This observation is consistent with a model where FBFs regulate the rates of SPC cell division and meiotic entry by affecting separate sets of target mRNAs.

In vitro and in vivo studies have revealed that FBFs recognize the same FBE 3'UTR motif and bind largely same mRNAs (*Bernstein et al., 2005*; *Prasad et al., 2016*; *Porter et al., 2019*). Based on this, we speculate that antagonistic regulation of FBF target mRNAs results from FBFs competing for the same transcripts. Alternatively, since some FBF targets have multiple FBEs, FBFs might co-bind the transcripts thus subjecting mRNAs to combinatorial regulation. Despite FBF localization to distinct RNA granules, diffuse cytoplasmic fraction of FBF proteins might be sufficient for such co-regulation. Either mechanism would result in FBF target mRNAs differentially responding to the presence of a single or both FBFs, and would explain how change-of-function mutants such as FBF-2vrm and FBF-1(FBF-2vr4) are able to rescue loss-of-function of the non-cognate *fbf* (*Table 4*).

**Table 4.** Variable regions outside of the RNA-binding domain regulate FBF function.

| Transgene | Mutated variable region (VR) sequence | Rescues *fbf-1(lf)*? | Rescues *fbf-2(lf)*? | Dependent on CCR4-NOT |
|---|---|---|---|---|
| GFP::FBF-2wt | N/A | No | Yes | No[*] |
| GFP::FBF-1wt | N/A | Yes | No | Yes[*] |
| GFP::FBF-2(vrm) | mutated VR1, 2; VR4 deleted | Yes | No | Yes[†] |
| GFP::FBF-1(FBF-2vr4) | VR4 swapped with FBF-2 | No | Yes | No[†] |
| GFP::FBF-1(FBF-2vr3) | VR3 swapped with FBF-2 | Yes | No | Yes[†] |

Rescue assays were performed by crossing transgenic GFP::FBFs into loss of function mutant of each *fbf*, followed by SPC zone length measurement (**Figure 6**, **Figure 6—figure supplement 1**, and **Figure 7—figure supplement 1**). Dependence on CCR4-NOT was defined as a decrease in the length of SPC zone after knocking down CCR4-NOT subunits.

[*] – analyzed in single *fbf* loss-of-function mutants, **Figure 3B**.

[†] – analyzed in the strains containing GFP::FBF transgenes in *fbf-1 fbf-2* double-mutant background, **Figures 6E** and **7D**.

## Regulation of cyclin B by PUF-family proteins in stem cells

PUF mRNA targets have been studied in multiple organisms including *C. elegans*, mouse, and human identifying thousands of target mRNAs (*Chen et al., 2012*; *Galgano et al., 2008*; *Kershner and Kimble, 2010*; *Morris et al., 2008*; *Porter et al., 2019*; *Prasad et al., 2016*). One highly conserved group of PUF regulatory targets is related to the control of cell cycle progression. In several developmental contexts, stem cells undergo rapid G1/S transitions and spend an extended time in G2, as observed for *C. elegans* germline stem cells as well as for mouse and human embryonic stem cells (*Fox et al., 2011*; *Lange and Calegari, 2010*; *Orford and Scadden, 2008*). Human, mouse, and *C. elegans* PUF proteins repress Cip/Kip family cyclin-dependent kinase inhibitors (*Kalchhauser et al., 2011*; *Kedde et al., 2010*; *Lin et al., 2019*). This repression was found to be important for cell cycle progression of human and mouse cells through G1 (*Kedde et al., 2010*; *Lin et al., 2019*). Additionally, mitotic cyclins B and A are among the core targets of PUF proteins across species including nematode FBFs (*Kershner and Kimble, 2010*; *Porter et al., 2019*; *Prasad et al., 2016*), *Drosophila* Pumilio (*Asaoka-Taguchi et al., 1999*), human and mouse PUM1 and PUM2 (*Chen et al., 2012*; *Galgano et al., 2008*; *Hafner et al., 2010*; *Morris et al., 2008*), and yeast Puf proteins (*Gerber et al., 2004*; *Wilinski et al., 2015*). Cyclin B regulation by PUFs contributes to cell cycle control of *Drosophila* embryonic cell divisions (*Asaoka-Taguchi et al., 1999*; *Vardy and Orr-Weaver, 2007*) and to the control of meiotic resumption during *Xenopus* and zebrafish oocyte maturation (*Kotani et al., 2013*; *Nakahata et al., 2003*; *Ota et al., 2011*). Here, we find that PUF-mediated regulation of mitotic cyclins affects cell cycle dynamics in the germline stem cells of *C. elegans*. A recent report suggests that regulation of cyclin B by PUFs is also observed in mouse embryonic stem cells (*Uyhazi et al., 2020*). It would be interesting to investigate whether PUF-family proteins might affect cell cycle of embryonic stem cells through control of cyclin B expression. The slow cell cycle phenotype of the *fbf-2(lf)* mutant SPCs was rescued by introduction of a CYB-2.1 transgene with mutated FBE elements (**Figure 2D**), suggesting that the levels of B-type cyclins are limiting for SPC progression through cell cycle in this genetic background. Among the B-type cyclins, CYB-3 has emerged as a major G2 regulator in *C. elegans* germline stem cells (*Lara-Gonzalez et al., 2019*). Since *cyb-3* lacks canonical FBE elements in its 3'UTR, testing whether FBFs might regulate CYB-3 levels to control SPC division rate will require identification of the relevant non-canonical binding sites.

## mRNA deadenylation and PUF-mediated repression

Multiple studies suggest that deadenylation contributes to PUF-mediated translational repression (*Goldstrohm et al., 2006*; *Kadyrova et al., 2007*; *Van Etten et al., 2012*; *Weidmann et al., 2014*). CCR4-NOT deadenylation machinery is conserved in evolution from yeast to humans (*Collart et al., 2017*; *Wahle and Winkler, 2013*). Although deadenylation is required for germline stem cell maintenance in flies, nematodes and mice (*Berthet et al., 2004*; *Fu et al., 2015*; *Joly et al., 2013*; *Nakamura et al., 2004*; *Nousch et al., 2019*; *Shan et al., 2017*; *Suh et al., 2009*), the contribution of deadenylation to PUF translational repression in vivo is still controversial (*Weidmann et al., 2014*). Previous studies of CCR4-NOT component CCF-1 in *C. elegans* suggested that paralogous

PUF proteins FBF-1 and FBF-2 might employ both CCF-1-dependent and CCF-1-independent regulatory modes (*Suh et al., 2009*). Here, we find that FBF-1 and FBF-2 differentially cooperate with CCR4-NOT deadenylation machinery in *C. elegans* germline SPCs (*Figure 3*).

Multiple lines of evidence suggest that FBF-1's function in vivo is supported by the CCR4-NOT deadenylation. First, the size of germline SPC zone maintained by FBF-1 in the absence of FBF-2 is significantly reduced by a knock-down of CCR4-NOT deadenylase components (*Figure 3B*). Second, FBF-1-mediated repression of FBF target reporter in vivo requires CCR4-NOT deadenylase (*Figure 3C,D*). By contrast, SPC zone maintained by FBF-2 and repression of reporter by FBF-2 in the absence of FBF-1 are not affected by CCR4-NOT component knock down. Taken together, these observations provide genetic evidence that CCR4-NOT promotes FBF-1 function in germline SPCs. The increase in FBF-1 protein levels that we observed after knocking down the CCR4-NOT subunit *ccf-1* (*Figure 3—figure supplement 1C*) might result from the relief of FBF-1 auto-regulation (*Lamont et al., 2004*). Third, both endogenous FBF-1 and GFP::FBF-1 colocalize with a core CCR4-NOT subunit 3xFLAG::CCF-1 in vivo by co-immunostaining (*Figure 4A,C*). Additionally, an in vivo test of protein interaction between GFP::FBF-1 and 3xFLAG::CCF-1 using proximity ligation assay detects positive signal suggesting that these proteins reside in the same complex (*Figure 4D*). By contrast, there's significantly less in vivo colocalization and proximity between GFP::FBF-2 and 3xFLAG::CCF-1. These data are consistent with the model that FBF-1 and FBF-2 form distinct RNP complexes, of which FBF-1 complex preferentially includes CCR4-NOT deadenylase. Finally, we assessed the FBF target poly(A) tail length in the nematodes mutant for each *fbf*, and found that the poly(A) tail length of FBF targets *cyb-2.1* and *htp*-1 was relatively shorter in *fbf-2(lf)* background than in wild type and *fbf-1(lf)*. By contrast, *fbf-1(lf)* mutation resulted in a shift of poly(A) tail distributions to longer lengths compared to the wild type (*Figure 5*). We conclude that FBF-1 selectively cooperates with deadenylation machinery to promote translational repression of target mRNAs (*Figure 8B*). Conversely, FBF-2 protects the target mRNAs from deadenylation and promotes repression of target mRNAs by a deadenylation-independent mechanism.

The two FBF proteins are 91% identical in primary sequence (*Zhang et al., 1997*). If FBFs have distinct abilities to engage deadenylation machinery, what are the features of FBF-2 that prevent it from cooperating with CCR4-NOT? PUF RNA-binding domain is sufficient for a direct interaction with the CCF-1 subunit of CCR4-NOT and its homologs in multiple species, including *C. elegans* (*Goldstrohm et al., 2006*; *Hook et al., 2007*; *Kadyrova et al., 2007*; *Suh et al., 2009*; *Van Etten et al., 2012*). However, protein sequences outside of the well-structured RNA-binding domain can promote PUF-induced deadenylation and are hypothesized to function either through improved recruitment of CCR4-NOT complex or through allosteric activation of CCR4-NOT (*Webster et al., 2019*). We find that the Variable Region (VR) sequences outside of the RNA-binding domain of FBF-1 and FBF-2 play a key role in determining whether these proteins are able to cooperate with CCR4-NOT (*Table 4*). Mutations of three VRs (VR1, 2, and 4) in FBF-2 result in a protein that now cooperates with CCR4-NOT, suggesting that these regions are necessary to prevent the wild-type FBF-2 from engaging with the deadenylase (*Figure 6E,F*). Conversely, swapping the VR4 of FBF-2 onto FBF-1 renders the chimeric protein FBF-1(FBF-2vr4) insensitive to CCR4-NOT knockdown, indicating that VR4 of FBF-2 is sufficient to prevent cooperation with CCR4-NOT (*Figure 7D*). By contrast, swapping VR3 residing within FBF-2 RNA-binding domain into FBF-1 does not affect the FBF-1(FBF-2vr3) chimera's cooperation with CCR4-NOT, supporting the importance of protein sequences outside of the RNA-binding domain for cooperation with CCR4-NOT. Overall, we conclude that the protein regions outside of the conserved PUF RNA-binding domain regulate the repressive action mediated by each PUF protein homolog. As a result, distinct sequences flanking the RNA-binding domain may lead to differential preference of regulatory mechanisms exerted by individual PUF-family proteins (*Figure 8B*). Identifying the sequences outside of FBF-1 RNA-binding domain that promote its cooperation with CCR4-NOT remains a subject for future studies.

In conclusion, our results suggest a new mechanism regulating stem cell mitotic activity in conjunction with meiotic entry rate or differentiation in the *C. elegans* germline through antagonistic regulation of key mRNA targets by PUF family FBF proteins. Complementary activities of FBF-1 and FBF-2 combine to fine tune SPC proliferation and meiotic entry coordinately regulating both processes. PUF proteins are conserved stem cell regulators in a variety of organisms, and their control of target mRNAs that affect proliferation and differentiation is widespread as well. The future

challenge will be to determine whether PUF-dependent RNA regulation in other stem cell systems might be modulated to adjust stem cell division rate along with changing the rate of differentiation.

# Materials and methods

## Key resources table

| Reagent type (species) or resource | Designation | Source or reference | Identifiers | Additional information |
|---|---|---|---|---|
| Strain, strain background (*C. elegans*) | JK3022 | *Crittenden et al., 2002* | | *fbf-1(ok91) II* |
| Strain, strain background (*C. elegans*) | JK3101 | *Lamont et al., 2004* | | *fbf-2(q738) II* |
| Strain, strain background (*C. elegans*) | GC833 | *Pepper et al., 2003a* | | *glp-1(ar202) III* |
| Strain, strain background (*C. elegans*) | JH3269 | *Putnam et al., 2019* | | *pgl-1::gfp(ax3122) IV* |
| Strain, strain background (*C. elegans*) | JH2766 | *Merritt and Seydoux, 2010* | | *axIs1922[Ppie-1::GFP:: H2B::htp-2 3'UTR]* |
| Strain, strain background (*C. elegans*) | UMT433 | This paper | | *mntSi33 [Pgld-1::3xFLAG:: CYB-2.1::cyb-2.1 3'UTR] unc-119(ed3) III* |
| Strain, strain background (*C. elegans*) | UMT394 | This paper | | *mntSi29 [Pgld-1::3xFLAG:: CYB-2.1::cyb-2.1 3'UTR (fbm)] unc-119(ed3) III* |
| Strain, strain background (*C. elegans*) | UMT360 | This paper | | *mntSi23 (Pgld-1::3xFLAG:: CCF-1::ccf-1 3'UTR) II; unc-119(ed3) III* |
| Strain, strain background (*C. elegans*) | UMT338 | This paper | | *mntSi21 [Pgld-1::patcGFP:: fbf-1 3'UTR] unc-119(ed3) III* |
| Strain, strain background (*C. elegans*) | UMT389 | This paper | | *mntSi28 [Pgld-1::patcGFP:: fbf-1::fbf-1 3'UTR] unc-119(ed3) III* |
| Strain, strain background (*C. elegans*) | UMT378 | This paper | | *mntSi27 [Pgld-1::patcGFP:: fbf-2::fbf-2 3'UTR] unc-119(ed3) III* |
| Strain, strain background (*C. elegans*) | UMT441 | This paper | | *mntSi32 [Pgld-1::patcGFP:: FBF-2vrm::fbf-2 3'UTR] unc-119(ed3) III* |
| Strain, strain background (*C. elegans*) | UMT373 | This paper | | *mntSi26 [Pgld-1::patcGFP:: fbf-1(fbf-2vr3)::fbf-1 3'UTR] unc-119(ed3) III* |
| Strain, strain background (*C. elegans*) | UMT411 | This paper | | *mntSi31 [Pgld-1::patcGFP:: fbf-1(fbf-2vr4)::fbf-1 3'UTR] unc-119(ed3) III* |
| Antibody | Mouse monoclonal anti-FLAG M2 | Sigma-Aldrich Cat# F1804 | RRID:AB_262044 | (1:1,000) |
| Antibody | Rabbit monoclonal anti-GFP | Thermo-Fisher Cat# G10362 | RRID:AB_2536526 | (1:200) |

*Continued on next page*

*Continued*

| Reagent type (species) or resource | Designation | Source or reference | Identifiers | Additional information |
|---|---|---|---|---|
| Antibody | Mouse monoclonal anti-phospho-Histone H3 pSer10 6G3 | Cell Signaling Technology Cat# 9706 | RRID:AB_331748 | (1:400) |
| Antibody | Rabbit polyclonal anti-REC-8 | Novus Biologicals Cat# 29470002 | RRID:AB_2178279 | (1:500) |
| Antibody | Mouse monoclonal anti-PGL-1 K76 | DSHB Cat# K76 | RRID:AB_531836 | (5.2 µg/ml) |
| Antibody | Rabbit polyclonal anti-FBF-1 | *Voronina et al., 2012*; PA2388 | | (3.5 µg/ml) |
| Antibody | Mouse monoclonal anti-Tubulin DM1A | Sigma-Aldrich Cat# T6199 | RRID:AB_477583 | (1:300) |
| Antibody | Goat anti-mouse IgG (H+L) 594 | Jackson ImmunoResearch Cat# 115-585-146 | RRID:AB_2338881 | (1:500) |
| Antibody | Goat anti-rabbit IgG (H+L) 594 | Jackson ImmunoResearch Cat# 111-585-144 | RRID:AB_2307325 | (1:500) |
| Antibody | Goat anti-rabbit IgG (H+L) 488 | Jackson ImmunoResearch Cat# 111-545-144 | RRID:AB_2338052 | (1:500) |
| Antibody | Goat anti-mouse IgM 594 | Jackson ImmunoResearch Cat# 115-585-020 | RRID:AB_2338874 | (1:500) |
| Antibody | Goat anti-mouse HRP | Jackson ImmunoResearch Cat# 115-035-003 | RRID:AB_10015289 | (1:5,000) |
| Antibody | Goat anti-rabbit HRP | Jackson ImmunoResearch Cat# 111-035-003 | RRID:AB_2313567 | (1:5,000) |
| Recombinant DNA reagent | Plasmid: control RNAi | *Timmons and Fire, 1998* | pL4440 | |
| Recombinant DNA reagent | Plasmid: *let-711* RNAi | Source BioScience Ahringer RNAi Collection | | |
| Recombinant DNA reagent | Plasmid: *ccr-4* RNAi | Source BioScience Ahringer RNAi Collection | | |
| Recombinant DNA reagent | Plasmid: *ccf-1* RNAi | *Sönnichsen et al., 2005* | | |
| Recombinant DNA reagent | Plasmid: *cyb-2.1* RNAi | This paper | | *cyb-2.1* genomic CDS in pL4440 |
| Sequence-based reagent | act-1.qF | *Merritt and Seydoux, 2010* | qPCR primers | GGCCCAATCCA AGAGAGGTATC |

*Continued on next page*

*Continued*

| Reagent type (species) or resource | Designation | Source or reference | Identifiers | Additional information |
|---|---|---|---|---|
| Sequence-based reagent | act-1.qR | *Merritt and Seydoux, 2010* | qPCR primers | CAACACGAAGCT CATTGTAGAAGG |
| Sequence-based reagent | unc-54.qF | This paper | qPCR primers | agagagcaggttttggaggat |
| Sequence-based reagent | unc-54.qR | This paper | qPCR primers | ttgagggtgacctcatttcc |
| Sequence-based reagent | him-3.qF | *Merritt and Seydoux, 2010* | qPCR primers | AGAGATTTTCGTATC TCTAAATAACGGAATC |
| Sequence-based reagent | him-3.qR | *Merritt and Seydoux, 2010* | qPCR primers | GGGTGTATAGTCTTT TGGAGCTTTTTC |
| Sequence-based reagent | htp-1.qF | *Merritt and Seydoux, 2010* | qPCR primers | ATTCGGAGGACA GTGACACAA |
| Sequence-based reagent | htp-1.qR | *Merritt and Seydoux, 2010* | qPCR primers | GTGCTTTCTCGAG AGACTCAGTTATATC |
| Sequence-based reagent | htp-2.qF | *Merritt and Seydoux, 2010* | qPCR primers | ATCGTTCAATTCG GAGGACAC |
| Sequence-based reagent | htp-2.qR | *Merritt and Seydoux, 2010* | qPCR primers | GTGTTTTCTCGA GAGAATCGGTTATATT |
| Sequence-based reagent | cyb-1.qF | This paper | qPCR primers | CCAACAACAGACGAACATCG |
| Sequence-based reagent | cyb-1.qR | This paper | qPCR primers | CTGGATTGGA TGGCTTGAGT |
| Sequence-based reagent | cyb-2.1.qF | This paper | qPCR primers | AAACCACGAAAAATGCCGT |
| Sequence-based reagent | cyb-2.1.qR | This paper | qPCR primers | TGAAGCTGTCGTCAAGAACA |
| Sequence-based reagent | cyb-2.2.qF | This paper | qPCR primers | CAAGAATCAACATGAAAACGGAT |
| Sequence-based reagent | cyb-2.2.qR | This paper | qPCR primers | TCAGCCATGCAATTGAACTC |
| Sequence-based reagent | cyb-3.qF | This paper | qPCR primers | ACACCATTCAGAAGCTTGCAT |
| Sequence-based reagent | cyb-3.qR | This paper | qPCR primers | AGCGATCTCCGGAAAGGTAG |
| Sequence-based reagent | cyb-2.1.PAT | This paper | PCR primers | tacgttcctgtgttctgctt |
| Sequence-based reagent | htp-1.PAT | This paper | PCR primers | tggcatgttctatatcatcgtgt |

## *C. elegans* culture and strains

All *C. elegans* hermaphrodite strains (*Supplementary file 1*) used in this study were cultured on NNGM plates seeded with OP50 as per standard protocols (*Brenner, 1974*). All GFP-tagged transgenic animals were cultured at 24°C to avoid GFP silencing. Temperature sensitive allele *glp-1(ar202)* is a gain-of-function (gf) mutant and is referred to as *glp-1(gf)* in this study. *glp-1(gf)* is fertile at 15° C, but sterile at 25°C and produces tumorous germlines. *glp-1(gf)* was crossed with each single *fbf* loss-of-function (lf) mutant, *fbf-1(ok91)* and *fbf-2(q738)*, to generate *fbf-1(lf); glp-1(gf)* and *fbf-2(lf)/ mIn1; glp-1(gf)*. Double mutant strains and *glp-1(gf)* single mutant were maintained at 15°C. Propagation of *fbf-2(lf); glp-1(gf)* for large-scale sample collection is detailed below. Synchronized L1 larvae of *glp-1(gf)* strains were cultured at 25°C until early adulthood. RNA was extracted from tumorous worms and was subsequently used for qPCR and poly(A) tail length analysis.

## Propagation of *fbf-2(q738); glp-1(ar202gf)* nematodes for bulk sample collection

Sterility of *fbf-2(lf); glp-1(gf)* nematodes was partially rescued to 50–66% fertility by a mild *glp-1 (RNAi)*. The cultures of *fbf-2(lf); glp-1(gf)* were propagated on *glp-1(RNAi)* bacteria at 15°C. Due to heritable RNAi, the rescued fertility was retained up to the third generation after transfer to *E. coli* food, OP50. To collect large amounts of *fbf-2(lf); glp-1(gf)* worms with tumorous germlines, the culture of *fbf-2; glp-1(ar202); glp-1(RNAi)* was expanded on OP50 plates at 15°C before synchronizing by bleach. Then, synchronous L1 larvae were cultured on OP50 at 25°C.

## Transgenic animals

All transgene constructs were generated by Gateway cloning (Thermo Fisher Scientific). GFP::FBF-1 and GFP::FBF-2 constructs were generated with the *gld-1* promoter, patcGFP containing introns (*Frøkjær-Jensen et al., 2016*), *fbf-1* or *fbf-2* genomic coding and 3'UTR sequences in pCFJ150 (*Frøkjaer-Jensen et al., 2008*). GFP::FBF-2(vrm) was generated with *gld-1* promoter, patcGFP, *fbf-2* genomic coding sequences with variable regions 1, 2, and four mutated (P28A, S136A, K137A, Δ139–140, Δ607–632; *Wang et al., 2016*), and *fbf-2* 3'UTR in pCFJ150. GFP::FBF-1(FBF-2vr4) and GFP::FBF-1(FBF-2vr3) constructs were generated with *gld-1* promoter, patcGFP, *fbf-1* genomic coding sequences with swapped variable regions 4 or three from *fbf-2*, and *fbf-1* 3'UTR sequences in pCFJ150. 3xFLAG::CCF-1 construct contains *gld-1* promoter, *ccf-1* genomic coding and 3' UTR sequences in pCFJ150. 3xFLAG::CYB-2.1wt and 3xFLAG::CYB-2.1fbm constructs contain *gld-1* promoter, *cyb-2.1* genomic coding and 3' UTR sequences with either wild type (wt, 5' UGUxxxAU 3') or mutated (fbm, 5' ACAxxxAU 3') FBF-binding sites in pCFJ150.

A single-copy insertion of each GFP-tagged FBF transgene and CYB-2.1 transgenes was generated by homologous recombination into universal *Mos1* insertion site on chromosome III after Cas9-induced double-stranded break (*Dickinson et al., 2013*; *Wang et al., 2016*). Similarly, single-copy insertion of 3xFLAG-tagged CCF-1 was generated by targeting universal *Mos1* insertion site on chromosome II. Transgene insertion into universal *Mos1* insertion sites was confirmed by PCR.

## Germline SPC zone measurement

*C. elegans* were synchronized by bleaching, and hatched L1 larvae were plated on NNGM plates with OP50 bacteria or RNAi culture, grown at specified temperatures and harvested at varying time points depending on the experiment. For the time course of SPC zone length, L1 larvae of *fbf-1(lf)*, *fbf-2(lf)* and the wild type (N2) were cultured at 24°C and dissected at 46 hr (early adults that have initiated oogenesis), 52 hr (adults) and 63 hr (older adults) time points. L1 larvae of *fbf-2(lf); cyb-2.1fbm*, *fbf-2(lf); cyb-2.1wt* and *fbf-2(lf)* were grown at 15°C for 5 days until adult stage. In all other SPC zone quantification assays, L1 larvae of all worm strains were cultured at 24°C and dissected for staining at 52 hr time point. Gonads were dissected and stained for mitotic marker REC-8 (*Hansen et al., 2004*), and the length of SPC zone in each germline was measured by counting the number of germ cell rows positive for REC-8 staining before transition zone, ending with the last row fully occupied by REC-8-positive cells. Measuring the extent of progenitor zone by counting the number of cell rows positive for mitotic marker REC-8 provides a reliable estimate of progenitor cell numbers and correlates with progenitor cell numbers in the key genotypes including wild type, *fbf-1 (lf)*, and *fbf-2(lf)* ($R2 = 0.779$; *Figure 1—figure supplement 2*).

## M phase index measurement

Synchronous cultures of wild type (N2), *fbf-1(lf)* and *fbf-2(lf)* L1 larvae were cultured at 24°C for 52 hr. Gonads were dissected and stained for a mitotic marker REC-8 and an M phase marker phospho-Histone H3 (pH3). Primary and secondary antibodies are described in *Supplementary file 2*. M phase index was calculated by dividing the number of pH3-positive SPCs by the number of REC-8-positive SPCs. Percent differences in mitotic indices were calculated as for G2-phase length or differentiation rate.

## EdU labeling

S-phase index, G2-phase length and differentiation rate of germ cells were measured by feeding *C. elegans* EdU-labeled bacteria for varying amounts of time at 24°C (*Crittenden et al., 2006*;

*Fox et al., 2011*; *Kocsisova et al., 2018*), with three or four replicates. EdU bacteria plates were prepared by diluting an overnight culture of thymine deficient MG1693 *E. coli* (The Coli Genetic Stock Center; Yale University) 1/25 in 1% glucose, 1 mM MgSO$_4$, 5 µg/mL thymine, 6 µM thymidine and 20 µM EdU in M9 minimal media. This culture was grown at 37°C for 24 hr, pelleted by centrifugation, resuspended in 10 mL M9 minimal media and plated on 60 mm NNGM plates. Worm strains were synchronized by bleaching, hatched overnight and L1 larvae were cultured on OP50 plates at 24°C for ~50 hr to reach young adult stage, when they were exposed to EdU-labeled bacteria. After feeding for specified time, worms were picked off EdU plates, dissected on poly-L-lysine treated slides, frozen on dry ice and fixed in ice-cold 100% methanol for 1 min followed by 2% paraformaldehyde/100 mM K$_2$HPO$_4$ pH 7.2 for 5 min. Next, slides were blocked in PBS/0.1% BSA/0.1% Tween-20 (PBS-T/BSA) for 30 min at room temperature. Samples were incubated with primary antibodies against either phospho-Histone H3 or REC-8 (*Supplementary file 2*). After overnight incubation with primary antibody slides were washed 3 × 10 min with PBS-T/BSA and incubated with secondary antibody for 1.5 hr at room temperature. Secondary antibodies were either Alexa Fluor 594-conjugated goat anti-mouse IgG (H+L) or Alexa Fluor 594-conjugated goat anti-rabbit IgG (H+L), respectively (*Supplementary file 2*). After incubation with secondary antibody slides were washed 4 × 15 min with PBS-T/BSA. Next, the Click-iT reaction was performed according to the manufacturer's instructions (Molecular Probes) with the exception that 2 × 30 min Click-iT reactions were performed to increase the signal of the Alexa 488 dye. After incubation with the second Click-iT reaction, slides were washed 4 × 15 min with PBS-T/BSA. Vectashield with DAPI (Vector Laboratories) was added to each sample before cover-slipping. Germline images were captured as z-stacks spanning the thickness of each germline using a Leica DM5500B microscope. For each replicate time point 7–14 germlines were scored and the data represent three or five biological replicates. Nuclei were manually counted using the Cell Counter plug-in in Fiji (*Schindelin et al., 2012*) and the Marks-to-Cells R script (*Seidel and Kimble, 2015*) was used to remove multiply-counted nuclei.

## S-phase index

Synchronous cultures of young adult nematodes were fed EdU-labeled bacteria for 30 min at 24°C. Germ cells were co-labeled with anti-REC-8 antibody. The S-phase index was calculated as the fraction of REC-8-positive cells that were also EdU-positive.

## G2 length and differentiation rate analysis

To calculate the **median duration of G2-phase** animals were fed EdU and collected at 30 min intervals from 0 to 3.5 hr. Germ cells were co-labeled with anti-pH3 antibody and the fraction of M-phase nuclei that have also completed G2-phase was determined by dividing the number of pH3 and EdU-positive nuclei by the total number of pH3 positive nuclei. The percent pH3 and EdU-positive nuclei was plotted on the y-axis against the duration of the EdU label on the x-axis and the data were fit to a sigmoidal varying slope curve using GraphPad Prism software, with top and bottom constrained at 100 and 0 respectively (*Figure 1—figure supplement 1B*). The $t_{50}$ value of the sigmoidal dose-response model was taken as the median duration of G2-phase, or the time at which 50% of pH3 positive cells are also EdU positive.

The **rate of meiotic entry** was calculated by feeding the worms EdU-labeled bacteria for 3, 6 or 10 hr and co-labeling the germ cells with anti-REC-8 antibody. The number of nuclei that had entered meiosis in the time since EdU exposure based on being REC-8 negative and EdU positive were counted for each time point. The number of nuclei that entered meiosis was plotted on the y-axis and the duration of the EdU label was plotted on the x-axis in GraphPad Prism software. Linear regression analysis was used to calculate the slope, which corresponds to the number of cells that have entered meiosis per hour. Percent differences in G2-phase length or differentiation rate were calculated through subtracting the mean value of median G2-phase length or differentiation rate of each *fbf* mutant from that of the wild type followed by dividing by the value of the wild type.

## Larval germ cell proliferation rate

Germ cell proliferation assays of *fbf* mutants were performed using strains where germ cells were identified by expression of CRISPR-tagged PGL-1::GFP. The nematodes were synchronized by bleaching and hatched L1s were fed on OP50 NNGM plates. The *pgl-1::gfp*, *fbf-1(ok91); pgl-1::gfp*

and *fbf-2(q738); pgl-1::gfp* strains were grown at 24℃. At 17 and 21 hr time points after the start of feeding samples were taken to image germ cell accumulation in L2 larvae. The data represent four biological replicates, and 15–21 germlines of each strain were scored per time point in each replicate. To analyze CYB-2.1 effect on larval germ cell proliferation, the *fbf-2(q738); pgl-1::gfp*, *fbf-2 (q738); cyb-2.1(wt); pgl-1::gfp*, and *fbf-2(q738); cyb-2.1(fbm); pgl-1::gfp* strains were grown at 15℃. At 41 and 46 hr time points after the start of feeding germ cells were imaged in L2 larvae, and the data represent five biological replicates with 13–20 germlines scored per time point in each replicate. The number of germ cells were scored in each germline by counting cells containing P granules. The doubling rate of larval germ cells was estimated by exponential fits performed independently for each biological replicate.

## RNAi treatment

The following RNAi constructs were used: *ccr-4*, *let-711* (**Kamath and Ahringer, 2003**), *ccf-1* (*cenix:341* c12; **Sönnichsen et al., 2005**) and *cyb-2.1* (genomic CDS) in pL4440 (**Timmons and Fire, 1998**). Empty vector pL4440 was used as a control in all RNAi experiments. All RNAi constructs were verified by sequencing. RNAi plates were prepared as previously described (**Wang et al., 2016**) and synchronously hatched L1 larvae were plated directly on RNAi plates, except for *let-711* and *ccf-1(RNAi)*, where L1 larvae were initially grown on OP50 plates and transferred to RNAi plates at the L3/L4 stage. For CCR4-NOT knockdown, L1 larvae were grown at 24℃ for 52 hr before analysis; for *cyb-2.1* knockdown, worms were grown at 15℃ for 120 hr before analysis. The effect of *cyb-2.1(RNAi)* was confirmed by western blot of 3xFLAG::CYB-2.1. The effectiveness of CCR4-NOT RNAi treatments was assessed by scoring sterility (**Figure 3—figure supplement 1**) or embryonic lethality (**Supplementary file 3**) in the F1 progeny of the fed animals.

RNA extraction and PAT-PCR *glp-1(gf)* and *fbf-1(lf); glp-1(gf) C. elegans* were synchronized using bleach, hatched L1s were cultured at 25℃ and worms were harvested after 52 hr. A subset of animals from each batch were dissected and germlines were stained with DAPI, anti-REC-8, and anti-phospho-histone H3 to assess abnormal proliferation. Although the documented phenotype of *glp-1 (ar202)* at 25℃ is ectopic germline proliferation in the proximal region, we have often observed formation of full germline tumors in all three genotypes in our culture conditions (**Figure 5—figure supplement 1A**). The RNA samples were prepared from the cultures with >77% full germline tumors. Worm pellets were washed twice with 1x M9 to remove OP50 bacteria, weighed, flash-frozen using dry ice/ethanol slurry and stored at −80℃. Three biological replicates were analyzed for each genetic background. Total RNA was isolated using Trizol (Invitrogen) and Monarch Total RNA miniprep kit (NEB). RNA concentration was measured using Qubit (Thermo Fisher). PAT-PCR for the FBF target *cyb-2.1* and control *tbb-2* was performed using a Poly(A) Tail-Length Assay Kit (Thermo Fisher). Briefly, G/I tailing, reverse transcription, PCR amplification and detection were performed following the kit protocol. Each G/I tailing reaction used 1 µg total RNA. During PCR amplification, 1 µl of diluted RT sample was used in each PCR reaction and a two-step PCR program was used: 94℃ for 2 min, (94℃ for 10 s, 60℃ for 1 min 30 s) x 35 cycles, 72℃ for 5 min. PCR products were assessed using 6 or 8% polyacrylamide gels (made with 29:1 Acrylamide/Bis Solution, Bio-Rad) electrophoresis. PCR products were visualized with SYBR Gold stain (Invitrogen) and recorded using ChemiDoc MP Imaging System (Bio-Rad). Poly(A) tail length distributions were compared using densitometry analysis in ImageJ following background subtraction and normalization. The peak locations were identified as maxima of average normalized intensity profiles.

## qPCR

The qPCR data represent three biological replicates of *glp-1(gf)*, four biological replicates of *fbf-2 (lf); glp-1(gf)*, and six biological replicates of *fbf-1(lf); glp-1(gf)*. cDNA was synthesized using the SuperScript IV reverse transcriptase (Thermo Fisher) using 2 µg RNA template per each 20 µg cDNA synthesis reaction. Quantitative PCR reactions were performed in technical triplicates per each input cDNA using iQ SYBR Green Supermix (Bio-Rad) with cDNA diluted 1:10 as template. Primers for *htp-1*, *htp-2*, *him-3*, and *act-1* were as described (**Merritt and Seydoux, 2010**). Primers for *cyb-1*, *cyb-2.1*, *cyb-2.2*, *cyb-3*, and *unc-54* were designed to span exon-exon boundaries to avoid amplification of residual genomic DNA. Abundance of each mRNA in two *fbf* mutants relative to the wild type was calculated using the comparative ΔΔCt method (**Pfaffl, 2001**) with actin *act-1* as a

reference gene. After the mRNA abundance of each tested gene was normalized to *act-1*, the fold change values from replicates were averaged. Finally, fold change values of each tested gene in *glp-1(gf); fbf-1(lf)* and *glp-1(gf); fbf-2(lf)* genetic backgrounds were scaled to the average value in *glp-1 (gf)* in which the mRNA abundance was set to 1. Differences in mRNA abundance between *glp-1(gf)* and *fbf-2(lf); glp-1(gf)* were evaluated by one-way ANOVA statistical tests with Student's t-test post-tests. Since all FBF targets analyzed in this paper are germline-enriched in the adult nematodes, the mRNA abundances in whole worm lysates reflect their abundances in the germline.

## Immunolocalization and image analysis

For all immunostaining experiments, *C. elegans* hermaphrodites were dissected and fixed as previously described (*Wang et al., 2016*). All primary antibody incubations were overnight at 4°C and all secondary antibody incubations were for 1.5 hr at room temperature. For colocalization analysis of endogenous FBF-1 and 3xFLAG::CCF-1, dissected gonads of *flag::ccf-1* were stained with anti-FBF-1 (Rabbit) and anti-FLAG primary antibodies (Mouse) (*Supplementary file 2*). For colocalization analysis of GFP::FBFs and 3xFLAG::CCF-1, dissected gonads of *3xflag::ccf-1; gfp::fbf-2* and *3xflag::ccf-1; gfp::fbf-1* were stained with rabbit anti-GFP and mouse anti-FLAG primary antibodies (*Supplementary file 2*). Secondary antibodies were Goat anti-Mouse or Goat anti-Rabbit. Germline images were acquired using Zeiss 880 confocal microscope. Localization of FBF granules relative to CCF-1 granules were analyzed in a single confocal section per germline with four to six germ cells in SPC zone by Pearson's correlation coefficient analysis using the JACoP plugin of ImageJ. For each worm strain, four to eight independent germline images were analyzed and Pearson's correlation coefficient values were averaged.

## Proximity ligation assay (PLA)

PLA was performed on dissected *C. elegans* gonads following a modified Duolink PLA Protocol as described (*Day et al., 2020*). Fixation was as previously described (*Wang et al., 2016*). Blocking steps included incubation in 1xPBS/0.1% Triton-X-100/0.1% BSA for 2 × 15 min at room temperature, in 10% normal goat serum for 1 hr at room temperature, and in Duolink blocking buffer for 1 hr at 37°C. Primary anti-GFP and anti-FLAG antibodies were diluted in Duolink diluent (*Supplementary file 2*). After overnight incubation with primary antibodies at 4°C, 1:5 dilutions of PLUS and MINUS Duolink PLA Probes were added to each slide and incubated at 37°C for 1 hr. Next, slides were incubated at 37°C for ligation (for 30 min) and amplification (for 100 min) steps and finally mounted with Duolink Mounting medium with DAPI. Images were acquired using Zeiss 880 confocal microscope. The ImageJ 'Analyze Particles' plugin was used to quantify PLA foci in germline images.

## FBF target reporter regulation assay

Reporter transgene with GFP fused to Histone H2B and the 3' untranslated region (UTR) of *htp-2* (*Merritt et al., 2008*; *Merritt and Seydoux, 2010*) was crossed into *rrf-1(lf), rrf-1(lf)/hT2; fbf-1(lf)* and *rrf-1(lf); fbf-2(lf)* genetic backgrounds. RNAi targeting *let-711* and *ccf-1* were conducted on these reporter strains as described above. The effectiveness of RNAi treatments was assessed by scoring F1 embryo lethality. RNAi-treated worms were dissected and fluorescent germline images were acquired on a Leica DFC300G camera attached to a Leica DM5500B microscope with a standard exposure. Percentage of germlines that exhibited target reporter derepression in the SPC zone was scored for each strain.

## Immunoblotting

Synchronous cultures of *C. elegans* were collected at the adult stage by washing in 1xM9 and centrifugation and worm pellets were lysed by sonication. Proteins from worm lysates were separated using SDS-PAGE gel electrophoresis and transferred to a 0.45 μm PVDF membrane (EMD Millipore) as previously described (*Ellenbecker et al., 2019*). Primary and secondary antibodies are described in *Supplementary file 2*. Blots were developed using Luminata Crescendo Western HRP substrate (EMD Millipore) and visualized using ChemiDoc MP Imaging System (Bio-Rad).

# Acknowledgements
We thank the members of Voronina laboratory for insightful discussions and Geraldine Seydoux for comments on our manuscript. We are grateful to Ella Baumgarten and Jessica Bailey for help with cloning and crosses. We appreciate Ariz Mohammad for sharing the modified R script (originally from the Kimble lab) and instructions on using R for cell counts. Some nematode strains used in this study were provided by the *Caenorhabditis* Genetics Center funded by the NIH (P40OD010440). Confocal microscopy was performed in the University of Montana BioSpectroscopy Core Research Laboratory operated with support from NIH awards P20GM103546 and S10OD021806. This work was supported by the NIH grants GM109053 to EV and P20GM103546 (S Sprang, PI; EV Pilot Project PI), American Heart Association Fellowship 18PRE34070028 to XW, and Montana Academy of Sciences award to XW.

# Additional information

## Funding

| Funder | Grant reference number | Author |
|---|---|---|
| National Institute of General Medical Sciences | GM109053 | Ekaterina Voronina |
| National Institute of General Medical Sciences | P20GM103546 | Ekaterina Voronina |
| American Heart Association | 18PRE34070028 | Xiaobo Wang |
| Montana Academy of Sciences | | Xiaobo Wang |

The funders had no role in study design, data collection and interpretation, or the decision to submit the work for publication.

## Author contributions
Xiaobo Wang, Conceptualization, Formal analysis, Funding acquisition, Investigation, Methodology, Writing - review and editing; Mary Ellenbecker, Formal analysis, Funding acquisition, Investigation, Methodology, Writing - review and editing; Benjamin Hickey, Formal analysis, Investigation; Nicholas J Day, Investigation, Methodology, Writing - review and editing; Emily Osterli, Mikaya Terzo, Investigation; Ekaterina Voronina, Conceptualization, Supervision, Funding acquisition, Investigation, Project administration

## Author ORCIDs
Xiaobo Wang [iD] https://orcid.org/0000-0003-3165-5109
Ekaterina Voronina [iD] https://orcid.org/0000-0002-0194-4260

## Decision letter and Author response
Decision letter https://doi.org/10.7554/eLife.52788.sa1
Author response https://doi.org/10.7554/eLife.52788.sa2

# Additional files

## Supplementary files
• Supplementary file 1. Nematode strains used in this study.

• Supplementary file 2. Antibodies used in the study.

• Supplementary file 3. Embryo lethality resulting from CCR4-NOT knockdown in the parent generation. Percent embryos failing to hatch was determined 24 hr after eggs were deposited on a plate. The data were obtained in two independent experiments.

• Transparent reporting form

## Data availability

All data generated or analysed during this study are included in the manuscript and supporting files.

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
