## [Decision Letter]

**Acceptance summary:**

This study provides a significant advance in our understanding of how stem cell population is regulated in *C. elegans* germline. It has been known that two PUF proteins FBF-1 and FBF-2 have redundant roles in this process, but there have been unexplained differences in their phenotypes and protein localization. This study highlights functional differences of FBF-1 and FBF-2, explaining how these two PUF proteins together regulate germline stem cell population. Specifically they showed that, although they target the same set of genes, their mechanisms of action is different: FBF-1 (but not FBF-2) relies on CCR4-NOT deadenylase for its function to down regulate target mRNA. Overall, the reviewers felt that this study provides an important advance in our understanding of how *C. elegans* germline stem cells are maintained, with generalizable knowledge that is likely broadly applicable to other systems.

**Decision letter after peer review:**

Thank you for submitting your article "PUF family proteins FBF-1 and FBF-2 regulate germline stem and progenitor cell homeostasis in *C. elegans*" for consideration by *eLife*. Your article has been reviewed by three peer reviewers, and the evaluation has been overseen by a Reviewing Editor and Marianne Bronner as the Senior Editor. The reviewers have opted to remain anonymous.

The reviewers have discussed the reviews with one another and the Reviewing Editor has drafted this decision to help you prepare a revised submission.

Summary:

This study provides a significant advancement in our understanding of how stem cell population is regulated in *C. elegans* germline. It has been known that two PUF proteins FBF-1 and FBF-2 have redundant roles in this process, but there have been unexplained differences in their phenotypes and protein localization. This study highlights functional differences of FBF-1 and FBF-2, explaining how these two PUF proteins together regulate germline stem cell population. Specifically they showed that, although they target the same set of genes, their mechanisms of action is different: FBF-1 (but not FBF-2) relies on CCR4-NOT deadenylase for its function to down regulate target mRNA.

Overall, the reviewers felt that this study provides an important advance in our understanding of how *C. elegans* germline stem cells are maintained, with generalizable knowledge that is likely broadly applicable to other systems. The reviewers provided suggestions to improve the accuracy/clarity of the manuscript.

Essential revisions:

1) The following is summary of required changes/revisions (suggested by multiple reviewers)

– Cell cycle phenotypes for fbf(-)s should be added to/improved

– RT-PCR data needs clarification (the tbb-2 negative control in particular and the statistical analysis), and potentially the addition of complementary CCR4-NOT data

– CYB-2.1 with mutated FBEs should include a wild-type control and more thorough characterization of the effect on cell cycle progression

– Deadenylation of CYB-2.1 needs additional controls (wild-type = *glp-1(gf)* alone) and to look at additional targets that might show a similar trend.

– While the specific suggestions differ, a little more from the domain swapping experiments would add to the manuscript – showing changes in co-localization, addressing the implications of their different rescue experiments, and/or experiments addressing readouts other than PZ/Rec8+ zone size.

– All think Figure 8 (the model) needs clarification/improvement to add to the manuscript

2) In addition, individual reviewers provided suggestions/comments, which can be addressed by textual changes (individual comments can be found at the bottom of this letter). Although these may not require additional experiments, we would like the authors to address these points by editing to improve accuracy/clarity of the manuscript.

Reviewer #1:

Wang et al. provide significant contributions to the interesting story of differing roles for two very similar PUF proteins FBF-1 and FBF-2 in the *C. elegans* germ line with implications for PUF proteins in general. First, based on cell cycle measurements, they define different and interesting roles for the FBFs in regulating rates of mitotic cell cycle progression and meiotic entry ("proliferation" and "differentiation"), and they identify a key cyclin. Second, they distinguish the role of FBF-1 as dependent on CCR4-NOT deadenylase, likely leading to degradation of target mRNAs. Third, they define sequences in FBF-1 that mediate CCR4-NOT dependence.

Comments and Suggestions:

1) Figure 5. A major conclusion is that FBF-1 is promoting deadenylation of its target mRNAs via CCR4-NOT. Demonstration of PolyA length differences with additional FBF-1 targets would bolster this conclusion.

2) mRNA quantification after CCR4-NOT RNAi as in and compared with Figure 1G would also further bolster the degradation conclusion.

3) Does the FBF-2 VR mutant now associate with CCF-1 in the proximal ligation assay? If so, this would strengthen the conclusion that the function has been shifted to FBF-1-CCR4-NOT.

4) Figure 8. Are you proposing a competition model? Can you clarify how your molecular model will play out in the control of the spatial extent of the PZ?

5) "Proliferation" and its control should be clarified where possible as control of "rate of mitotic cell cycle progression". In the past, there has been confusion in the field regarding "proliferation" as the alternative cell fate to "differentiation". For "meiotic entry", clarify where possible spatial position versus rate. Again, to help with the past confusion in the field, several places in the manuscript could clarify by substituting "differentiation" with position or rate of "meiotic entry".

6) Several comments regarding the term "SPC zone size", throughout the manuscript:

First, best to call the pool of REC-8-positive cells "progenitor cells", and the zone as the "progenitor zone", abbreviating as "PZ", rather than defining a new term and acronym.

Second, the "zone size" should be (1) standardized to numbers of cells in the key genotypes to account for possible differences in gonad width and (2) defined more clearly as a distance measurement since the term "size" has been used in the literature to mean cell numbers, and (3) justified better as a readout.

Third, since this distance measurement is a key phenotypic readout throughout the manuscript, define better how the position of the dotted lines indicating the edge of the PZ was determined: the distance between the end of the REC-8 positive cells and the dotted line is not uniform in the figures (compare, for example Figure 1C and 6B). It is especially unclear for Figure 7B.

7) Introduction paragraph two. The position of cell divisions is not uniform (e.g., Maciejowski et al.). Please restate.

8) Subsection “FBF-1 and FBF-2 differentially regulate mRNA abundance of target genes controlling proliferation and differentiation”. Early adult *glp-1(ar202)* shifted to 25º as L1 does not contain "only mitotic cells" (Pepper et al., 2003). Please restate.

9) I found the naming of the vr mutants and swaps made this section hard to follow. Perhaps you could name them based on the donor and target proteins?

10) Discussion paragraph two. This point is subtle and a bit confusing. Perhaps clarify that you mean the proportions of cycling-competent versus non-cycling-competent pre-meiotic S that normally contribute to the PZ?

11) In the same paragraph add "specific", "no specific molecular markers…"

Reviewer #2:

Wang et al. have proposed that FBF proteins function differentially in regulating the same set of mRNA targets. Using a set of FBF targets they demonstrated that mRNA is downregulated by FBF-1 but not FBF-2. They provided evidence that FBF-1, but not FBF-2, functions together with CCR-4/NOT to de-adenylated cyb-2.1 and possibly other FBF targets. They argued that 3 variable regions (VR), present in FBF-2, are critical for their distinct molecular identity, removing and swapping these regions would make them each behave like other. This work furthers the mechanistic understanding on how puf proteins function in regulating their targets, the logic is straightforward and the manuscripts clearly deliver the above findings to readers. However, the data provided by the Wang et al. suffers from improper statistical analysis, imprecise interpretation of result and needs additional data to support the claims in the manuscript.

1) FBF-1 and FBF-2 differentially modulate proliferation and meiotic entry of *C. elegans* germline SPCs.

The authors have found that fbf mutants have differential mitotic index and concluded that they have different rate of proliferation. For the *fbf-2* mutant, a longer cell cycle is supported by slightly longer G2 phase. However, for the *fbf-1* mutant G2 length is the same as wild type, which is inconsitent with the higher mitotic index (Lara-Gonzalez et al., 2019). Higher M-phase index could instead be due to cells spending more time in M-phase (which is possible due to overexpression of cell cycle regulators such as cyclin B proteins, which is downregulated to trigger M-phase exit). Additional data will need to be presented before making the conclusion that the *fbf-1* mutant has a shorter cell cycle. Determining the S-phase index could provide support for a shorter cell cycle. The S-phase index data could also bolster the point that *fbf-2* has a longer cell cycle, as the G2 lengthening is very modest and thus unlikely to explain the decreased meiotic output.

Importantly, the result that meiotic output of *fbf-1* and *fbf-2* mutants is higher and lower, respectively, than wild type is sufficient to explain the differences in the SPC zone length and FBF proteins role in meiotic entry. However, the data suffered from low sample size.

2) FBF-1 and FBF-2 differentially regulate mRNA abundance of target genes controlling proliferation and differentiation.

The authors hypothesized that "FBF-1 and FBF-2 might be acting differentially on the same set of mRNAs". They chose a set of mRNA targets of FBF (3 meiotic genes and 4 cell cycle regulators). Based on qPCR in fbf mutants and wild type (all in *glp-1(gf)* background, Figure 1G), authors argue that FBF-1 destabilize these target mRNAs, while FBF-2 promotes their accumulation. This data lays the foundation for their subsequent analysis. However, the data need improvement in several ways:

i) The authors have chosen linear trend analysis to show that, for all the mRNA tested, levels are higher in *fbf-1(-)* > wild type > *fbf-2*. While trend looks to be true, but given that control, tbb-2 mRNA levels are reduced by almost half in *fbf-2* mutant compared to wild type, with a possible exception cyb-2.2, none of the mRNAs tested can be different in *fbf-2* mutant compared to wild type. Based on the graph one can only argue that FBF-1 destabilize mRNA levels but FBF-2 does not have any appreciable effect.

ii) The statistical labels in Figure 1G is inconsistent with the statement made in the result. The authors claimed that other than cyb-1, all the other targets mRNA expression are increased in *fbf-1(-)*, but reduced in *fbf-2(-)*. While in Figure 1G, the statistical significance label for htp-2 and cyb-1 are EXACTLY the same. In addition, the authors are encouraged to explain why one-way ANOVA was chosen for their statistical analysis in the Materials and methods section. A pairwise comparison between wild type and each fbf single mutant should be performed for all the mRNAs tested; if there is a significant difference then one can make a conclusion.

iii) The authors indicate that they are studying these targets' regulation in germline tumor, however the RNA is from whole worms. The authors are encouraged to explain their approach.

3) Repression of cyclin B by FBF limits accumulation of germline SPCs.

The authors have mutated all three FBEs in cyb-2.1 and have successfully shown that CYB-2.1 protein levels have become higher in immunoblot. They also show that this mutation causes an increase of SPC size in *fbf-2* mutant and this can be reverted by depleting cyb-2.1. Based on this the authors conclude that "levels of B-type cyclins limit SPC proliferation rate in fbf-2(lf) and disruption of FBF-1-mediated repression of a single cyclin B gene is sufficient to affect the size of germline SPC zone".

There are multiple issues with this result:

i) *C. elegans* has ~12 puf genes (Prasad et al., 2016) and most of them have not been fully characterized; thus it is possible that mutated FBEs might be recognized by one or more of these puf proteins rather than FBF. Immunostaining and quantification of CYB-2.1 in extruded germlines of the *fbf-1* mutant and the *fbf-2* mutant would be required to claim that there is reciprocal regulation by the FBF proteins, rather than other puf proteins.

ii) Mutating FBEs in cyb-2.1, results in its higher expression. Rationale for checking the effect of its overexpression in fbf-2(-) background, instead of wild type, is not clear. Since there are no FBEs in cyb-2.1fbm, its effects are expected to be fbf-agnostic.

iii) Subsection “Repression of cyclin B by FBF limits accumulation of germline SPCs” paragraph three authors have concluded that increase in SPC zone size in fbf-2; cyb-2.1fbm vs *fbf-2* is due to increased proliferation and also the meiotic entry rate is unchanged between two mutants. They have not provided any evidence for either of these two possibilities, i.e. proliferation rate has increased in fbf-2; cyb-2.1fbm and meiotic entry rate is same in fbf-2; cyb-2.1fbm as compared to fbf-2.

4) FBF-1 promotes deadenylation of its target mRNA.

The authors moved back to cyb-2.1 to show that poly A tails are affected in fbf mutants without giving any rationale. The assay should be performed on a few more genes if any generalized statement statement is made, otherwise it must be restricted to cyb-2.1. Also wild type *(glp-1(gf)*) is missing in the assay, which makes it hard to interpret the data. Is FBF-1 alone or both FBFs responsible for deadenylation of cyb-2.1 mRNA or if FBF-1 and FBF-2 have opposing effect on cyb-2.1 mRNA?

5) While the authors present some solid lines of evidence that FBF-1 and FBF-2 are different (meiotic entry rate, FBF-1 acts with CCR-4/NOT to de-adenylated cyb-2.1 and there are specific sequences in FBF-2 that block this interaction), they are clearly redundant with each other for the stem cell fate and repression of htp-2. Thus, Figure 8 would be more useful to the community if their redundant function was also included.

6) FBF-1 and FBF-2 do not explain the spatial control of expression in the SPC zone and thus balance between proliferation and differentation, which is through *glp-1* targets lst-1 and sygl-1. It is surprising that there is no mention of lst-1 and sygl-1-1 in the Discussion, particularly since they are co-factors of FBF-1 and FBF-2 (Haupt et al., 2019, Shin et al., 2017).

Reviewer #3:

The *C. elegans* germline is a well-established model for investigating how signaling from a niche regulates the balance between stem cell self-renewal and differentiation. In the germline, the stem cell population is maintained as a pool, i.e. rather than undergoing invariant asymmetric divisions that always produce one stem cell and one differentiating daughter cell, the proliferative potential of any stem cell is determined by its distance from the niche and/or the level of niche signaling that it receives. The germline niche produces a Notch ligand which signals via the Notch receptor GLP-1 to promote the stem or mitotic cell fate. Once the level of Notch signaling drops below a threshold, cells enter meiosis. Downstream of Notch signaling, the FBF proteins, FBF-1 and FBF-2, act to maintain the stem cell pool and in *fbf-1*, *fbf-2* double mutants, at least at the adult stage, the stem cell population is lost and the germline resembles one in which Notch signaling has been removed entirely (Crittenden et al., 2002). This phenotype, in combination with the observation that FBF-1/2 share 89% identity and the majority of mRNA targets, has led to a simplified model in which they serve largely redundant roles. However, a fully redundant function is not supported by observations that single loss of function *fbf-1* or *fbf-2* germlines have distinct phenotypes (Lamont et al., 2004) and that FBF-1 and FBF-2 proteins have different localization patterns (Voronina et al., 2012).

Here the authors attempt to dissect what underlies this difference. Their work addresses two fundamental questions of broad impact/interest (1) how does niche signaling maintain a balance of proliferation and differentiation in a stem cell pool and (2) how can different modes of action be specified in largely similar proteins. As the first is broadly relevant to the stem cell field and the second is addressed within the context of two highly conserved proteins that play a role in stem cell maintenance in many organisms, this work is likely to be interesting to a wide audience. It should also be noted that because of the complexity of the system (the large number of FBF-1/2 targets and the number of other factors that also impinge upon stem cell dynamics), these questions are quite hard to address, and I commend the authors for trying to do so.

My critique of this manuscript can be broken into two categories: (1) it would benefit from some changes in writing to increase clarity and accessibility for non-experts and (2) there are a few questions that I had regarding experimental issues, some related to interpretation, which could be addressed in the text (points D, and F-H below) and some related to additional controls/experiments that would substantiate the authors' main conclusions (points A-C, E, and F). The suggested experiments are not substantially outside of those that the authors have already conducted and I think that most of them (enough to satisfy the majority of my major comments) could be performed within a reasonable time frame.

Major comments:

1) Writing/interpretation:

As someone who works in the germline, but not on this pathway in particular, I found that this paper was often hard to digest. It would be very helpful if the authors could make sure that their rationale for each experiment and its interpretation is accessible to a reader not intimately familiar with the field. For example, the difference in proliferative zone (PZ) size in *fbf-1(lf)* and *fbf-2(lf)*, I think pointing out here how a change in PZ size shows a shift in the balance between proliferation and differentiation, as if both were slowed down or sped up in complete coordination, the size of PZ would likely be constant. In addition, explaining why experiments were designed in a certain way and why the (perhaps) easier experiment is not feasible would be very helpful.

I also believe that the authors need to revise their language in several instances to avoid conflating two different questions/fields. For example, I found the use of "differentiation demands" confusing, because, for me, "differentiation demands" implies reproductive demand, i.e. the need for mature gametes. In both the *C. elegans* germline (e.g. Narbonne et al., 2015) and in other stem cell populations (e.g. Hsu, Li and Fuchs, 2014), signaling from differentiated/differentiating progeny to their stem cell parents plays a significant role in regulating stem cell proliferation, such that proliferative output matches the demand for differentiated progeny. I do not think that the authors are addressing this type of "feedback" signaling – i.e. they do not show that the rate of meiotic entry regulates the rate of proliferation, but rather that these two things are coordinated. I think that it is important to clarify their questiion to avoid confusion and/or erroneous expectations from people, such as myself, who are more used to thinking about feedback mechanisms. There are several instances of this throughout, including: "Complementary activities of FBF-1 and FBF-2 combine to fine tune SPC proliferation and differentiation to respond to proliferative demands of the tissue."

2) Experimental issues:

A) Figure 1. The cell cycle phenotype of *fbf-1(lf)* is puzzling. Since there are no quiescent cells in the germline, an increase in mitotic index (MI) ought to reflect either a faster passage through the mitotic cell cycle or a longer duration of mitosis. The authors report no change in the length of G2, which leads me to wonder whether S phase is accelerated or mitosis is delayed (I believe there are assays to assess both (S phase, Fox and Schedl, 2011) and M phase (Gerhold et al., 2015)). This result is also odd in light of their observation that the transcript levels of most of the Cyclin Bs (except for CYB-1), are highly elevated – i.e. why is the duration of G2 unaffected despite higher levels? The authors explanation in the discussion that a higher MI could be due to a smaller proliferative population, due to accelerated meiotic entry, would predict (I think) that the stem cell pool should eventually be lost – i.e. in older *fbf-1(lf)* adults, do they see a progressive loss of proliferative cells? As this interpretation (i.e. FBF-2, rather than FBF-1, being primarily responsible for cell cycle regulation) is key to their model figure at the end, it bears a closer look.

B) Also Figure 1. I am unsure as to how to interpret the RT-qPCR data. The authors state that their negative control (tbb-2) is unaffected, yet in fbf-2(lf) transcript levels appear to be at least a strongly depressed as those of the genes that they argue are maintained by FBF-2. It is possible that I am missing something here, but it would be good if the authors could address/explain this, as it is visually striking/distracting in their data/graph and if it is accurate, i.e. that in fbf-2(lf) transcript levels are generally lower, it is problematic for the interpretation that FBF-1 and FBF-2 have reciprocal effects on shared target transcripts. It is also not entirely clear to me what the linear trend test adds to the data interpretation, some clarification/discussion would be helpful.

C) In Figure 2, the authors address the hypothesis that slow G2 and low MI in fbf-2(lf) is due to FBF-1-mediated translational repression of cyb-2.1. The authors state that protein levels for CYB-2.1 with mutated FBF binding elements (FBEs) are elevated, but they cannot detect protein levels for the WT version, which could imply that it is not expressed at all. It would be helpful if the authors could should that both transgenes are expressed by looking at transcript levels. I also wonder if this result predicts that in wildtype germlines, expression of cyb-2.1 with mutated FBEs, but not wild-type cyb-2.1 would affect cell cycle progression? Finally, this hypothesis also seems to predict that a partial RNAi of FBF-1 in a fbf-2(lf) background, should rescue, at least to some extent, the fbf-2(lf) phenotype. If this experiment is technically feasible, it would greatly strength the authors argument.

D) In Figure 3, the authors use a translational reporter for the htp-2 3'UTR, which their RT-qPCR results suggest should be derepressed in *fbf-1(lf)*, but they do not seem to see this and I am unsure as to why not? In addition, their results strongly suggest that the change in PZ size in fbf-2(lf) requires CCR4-NOT complex members, which supports the hypothesis that FBF-1 activity with CCR4-NOT is leading to transcript loss. However, if FBF-1 acts with CCR4-NOT to regulate transcript levels, shouldn't RNAi of CCR4-NOT complex members phenocopy *fbf-1(lf)*? It would be helpful if the authors could address these questions. This may be another instance of where the data/results are not fully accessible to a non-expert.

E) In Figure 5, the authors quantify the polyA tail length of cyb-2.1 and show that it is shorter in fbf-2(lf), suggesting that it is being deadenylated, presumably by FBF-1/CCR4-NOT. It would strengthen the authors conclusion substantially, if they could show that the change in polyA tail length was rescued by depletion of CCR4-NOT complex members. It is also unclear to me why the authors compared *fbf-1(lf)* to fbf-2(lf) rather than to wildtype? I also wonder whether tbb-2 is the best control as the detection levels seem very low which could make slight changes in peak detection challenging.

F) Figure 6: I think that this a key place where the authors interpretation could be clarified somewhat. I believe that in Figure 6, the fact that FBF-2 without variable regions (VRs) 1, 2 and 4 cannot fully rescue the fbf-2(lf) phenotype in either a fbf-2(lf) or a *fbf-1(lf) fbf-2(lf)* background suggests that either this version of FBF-2 is only partially functional or it is acting like FBF-1. The authors then provide support for the latter interpretation by showing that the activity of FBF-2(vrm) requires CCR4-NOT and can also partially rescue *fbf-1(lf).* This last piece of data is in Figure 6—figure supplement 1, but, in my opinion, should be moved into the main figure panel. This also suggests that FBF-1 may be able to act with CCR4-NOT without any of its own variable regions, which (I think?) is somewhat surprising. The authors could test this by using a similar approach.

G) Figure 7: For the domain swapping experiments here, I believe the main conclusions are that VR3 (in the RNA-binding domain) is not required for FBF-1 activity, and that VR4 from FBF-2 is sufficient to prevent FBF-1 from acting with CCR4-NOT to regulate polyadenylation, as FBF-1 with FBF-2's VR4 cannot rescue *fbf-1(lf)* in a *fbf-1(lf)* or a *fbf-1(lf),fbf-2(lf)* background, while FBF-1 with FBF-2's R3 can. However, what I do not understand is why FBF-1 with FBF-2's VR4 rescues *fbf-2(lf)* (Figure 7—figure supplement 1)? Does this mean that it is acting as a dominant negative, since presumably wild-type FBF-1 is still present in this background?

H) Figure 8: The model figure as currently drawn doesn't really help me understand the authors' mains conclusions/findings. It would be great if this could be re-worked to reflect the effects of FBF-1 and FBF-2 on the stem cell population (i.e. perhaps in relation to PZ size which is their phenotypic readout for most experiments) and how these activities lead to a balance between stem cell maintenance and differentiation.

---

## [Author Response]

Essential revisions:1) The following is summary of required changes/revisions (suggested by multiple reviewers).– Cell cycle phenotypes for fbf(-)s should be added to/improved

As suggested by the reviewers, we now report new experiments to provide additional analysis of cell cycle in the three genotypes, including the S-phase index (Figure 1—figure supplement 1D). We additionally have determined cell doubling rate during larval development (as in Lara-Gonzalez et al., 2019) and found that the cell doubling rates in *fbf-1* mutant and wild type are similar, while the doubling time of *fbf-2* mutant is significantly longer (Figure 1E and Figure 1— figure supplement 1E, F).

– RT-PCR data needs clarification (the tbb-2 negative control in particular and the statistical analysis), and potentially the addition of complementary CCR4-NOT data

In response to the reviewers’ suggestions we have made the following changes:

1) We have replaced the negative control with *unc-54* myosin, and report that its abundance is not affected similarly to the FBF target abundance.

2) Upon repeating the qPCR experiments, we found that the increase of FBF target transcript levels in *fbf-1(lf)* was not reproducible. We have increased the number of biological replicates analyzed for each *fbf* mutant to gain confidence in the conclusions. The decrease in abundance of FBF targets in the *fbf-2(lf)* mutant background has been validated.

3) For statistical analysis, we have replaced linear trend analysis by a comparison between *fbf-2* mutant and the wild type background, since the target levels in *fbf-1(lf)* did not appear to change appreciably. This analysis revealed significant downregulation of multiple target mRNAs in *fbf-2(lf)* mutant (Figure 5—figure supplement 1B).

While we agree that it would be informative to test whether the decrease in FBF targets in *fbf-2(lf)* background was dependent on CCR4-NOT, we were not able to perform the analysis due to technical difficulties of growing large-scale cultures of *fbf-2(lf); glp-1(gf)* animals while achieving effective RNAi knockdown of CCR4-NOT.

– CYB-2.1 with mutated FBEs should include a wild-type control and more thorough characterization of the effect on cell cycle progression

We have included the following new experiments addressing these suggestions:

1) We have generated a new wild type *cyb-2.1* control transgene, and now document its expression at the mRNA and protein levels (Figure 2B, C).

2) We now include estimation of larval SPC proliferation rate comparing *fbf-2(lf)* with the strains expressing wild type or mutant *3xflag::cyb-2.1* transgenes. These new experiments suggest that cell cycle rate is accelerated in *fbf-2(lf); 3xflag::cyb-2.1fbm* compared to *fbf-2(lf)*, but not significantly changed in *fbf-2(lf); 3xflag::cyb-2.1wt* (Figure 2D).

3) We have documented meiotic entry in the *fbf-2(lf)* compared to the strains expressing wild type or mutant *3xflag::cyb-2.1* transgenes. These new experiments suggest that meiotic entry rate is not affected by the expression of *3xflag::cyb-2.1* transgenes (Figure 2E).

– Deadenylation of CYB-2.1 needs additional controls (wild-type = glp-1(gf) alone) and to look at additional targets that might show a similar trend.

We have addressed these suggestions by the following modifications of the experiment reflected in the updated Figure 5:

1) We have analyzed an additional FBF target, *htp-1*, and now find that its poly(A) tail similarly shortens in *fbf-2(lf)* and lengthens in *fbf-1(lf)* (Figure 5B, E)

2) We have included *glp-1(gf)* control to compare with *fbf* mutants, and find that the poly(A) tails of FBF targets in *glp-1(gf)* appear to be of intermediate lengths, in between the distributions in *fbf-1(lf)* and *fbf-2(lf)*.

3) As suggested by reviewer 3, we have replaced *tbb-2* by a different control, myosin heavy chain *unc-54* (Figure 5C, F)

– While the specific suggestions differ, a little more from the domain swapping experiments would add to the manuscript – showing changes in co-localization, addressing the implications of their different rescue experiments, and/or experiments addressing readouts other than PZ/Rec8+ zone size.

We have addressed these reviewer suggestions as follows:

1) As suggested by reviewer 1, we have tested whether FBF-2(VRM) variable region mutant associates with CCF-1. We found that proximity ligation assay detects significant enrichment of signal over background suggesting that FBF-2vrm RNP now incorporates CCR4-NOT deadenylase. These data are reported in Figure 6F and Table 3.

2) As suggested by reviewer 3, we moved *fbf-1* rescue experiment to the main Figure 6 (panel D).

3) As suggested by reviewer 3, we elaborated the discussion of rescue experiments (in the Discussion section).

– All think Figure 8 (the model) needs clarification/improvement to add to the manuscript.

In response to reviewers’ suggestions, we have modified the model as follows: We propose that FBFs bind to the same mRNAs, and likely to the same sites within these transcripts (as documented by multiple CLIP studies). We note in the Discussion that it is possible that FBFs either compete for the same binding sites or simultaneously bind to the transcripts that contain multiple FBEs. Either mechanism would explain how change-of-function mutants such as FBF-2vrm and FBF-1(FBF-2vr4) are able to rescue the loss-of-function of the non-cognate *fbf*.

The model in Figure 8 now emphasizes that both FBFs repress the bound target mRNAs and both FBFs can maintain SPCs on their own (thus highlighting their common, redundant function). Furthermore, we propose that FBF effects on the extent of the mitotic zone result from their regulation of stem cell dynamics (how fast the stem cells are able to divide and the rate of meiotic entry).

Reviewer #1:Wang et al. provide significant contributions to the interesting story of differing roles for two very similar PUF proteins FBF-1 and FBF-2 in the *C. elegans* germ line with implications for PUF proteins in general. First, based on cell cycle measurements, they define different and interesting roles for the FBFs in regulating rates of mitotic cell cycle progression and meiotic entry ("proliferation" and "differentiation"), and they identify a key cyclin. Second, they distinguish the role of FBF-1 as dependent on CCR4-NOT deadenylase, likely leading to degradation of target mRNAs. Third, they define sequences in FBF-1 that mediate CCR4-NOT dependence.Comments and Suggestions:1) Figure 5. A major conclusion is that FBF-1 is promoting deadenylation of its target mRNAs via CCR4-NOT. Demonstration of PolyA length differences with additional FBF-1 targets would bolster this conclusion.

As suggested, we now report poly(A) length differences for an additional FBF target, *htp-1* (Figure 5B, E).

2) mRNA quantification after CCR4-NOT RNAi as in and compared with Figure 1G would also further bolster the degradation conclusion.

Regretfully, we did not have time to optimize worm culture/RNAi to run this experiment.

3) Does the FBF-2 VR mutant now associate with CCF-1 in the proximal ligation assay? If so, this would strengthen the conclusion that the function has been shifted to FBF-1-CCR4-NOT.

As suggested, we have tested association between FBF-2(VRM) and CCF-1 and found that proximity ligation assay detects significant increase of signal over background, supporting the conclusion that FBF-2(VRM) is associating with CCR4-NOT deadenylase. These data are reported in Figure 6F and Table 3.

4) Figure 8. Are you proposing a competition model? Can you clarify how your molecular model will play out in the control of the spatial extent of the PZ?

As noted in the Essential Revisions, we speculate that FBFs compete for mRNAs targets. FBF binding to the same targets is documented by multiple CLIP studies. We note in the Discussion that it is possible that FBFs either compete for the same binding sites or simultaneously bind to the transcripts that contain multiple FBEs. Simultaneous binding is less likely given distinct localization of FBFs in vivo. Either mechanism would explain how change-of-function mutants such as FBF-2vrm and FBF-1(FBF-2vr4) are able to rescue the loss-of-function of the non-cognate *fbf*.

Our model further suggests that FBF effects on the extent of the mitotic zone result from their regulation of stem cell dynamics (how fast the stem cells are able to divide and the rate of meiotic entry).

5) "Proliferation" and its control should be clarified where possible as control of "rate of mitotic cell cycle progression". In the past, there has been confusion in the field regarding "proliferation" as the alternative cell fate to "differentiation". For "meiotic entry", clarify where possible spatial position versus rate. Again, to help with the past confusion in the field, several places in the manuscript could clarify by substituting "differentiation" with position or rate of "meiotic entry".

As suggested, we have replaced “proliferation” with “cell division rate” or “cell cycle progression rate” where appropriate. We have also clarified where “differentiation” was used to mean the rate of meiotic entry.

6) Several comments regarding the term "SPC zone size", throughout the manuscript:First, best to call the pool of REC-8-positive cells "progenitor cells", and the zone as the "progenitor zone", abbreviating as "PZ", rather than defining a new term and acronym.

In our manuscript, we decided to follow the lead of multiple preceding publications (for example, Korta et al., Development 2012; Gerhold et al., Curr Biol 2015; Roy et al., Dev Bio, 2016) in referring to the proliferative cells as “stem and progenitor cells” to reflect that this population likely includes both self-renewing stem cells and progenitor cells primed for differentiation.

Second, the "zone size" should be (1) standardized to numbers of cells in the key genotypes to account for possible differences in gonad width and (2) defined more clearly as a distance measurement since the term "size" has been used in the literature to mean cell numbers, and (3) justified better as a readout.

1) and 3) we have determined that progenitor zone size robustly correlates with the numbers of cells in the wild type and *fbf* single mutant germlines and now report this correlation in the Materials and methods section and include the data in the supplement (R^2^=0.7794; Figure 1—figure supplement 2).

2) We have replaced SPC zone size with either “length” or “extent” to clarify our metric.

Third, since this distance measurement is a key phenotypic readout throughout the manuscript, define better how the position of the dotted lines indicating the edge of the PZ was determined: the distance between the end of the REC-8 positive cells and the dotted line is not uniform in the figures (compare, for example Figure 1C and 6B). It is especially unclear for Figure 7B.

Thank you for bringing this to our attention. It appears that the differences between panels in Figures 1C, 6B and 7B were due to inconsistent cropping of the images and post-acquisition adjustment of anti-REC-8 staining intensity. These have been now corrected. We define boundary of the mitotic zone as the proximal-most full row of nuclei strongly positive for REC-8; this definition is now included in Materials and methods section.

7) Introduction paragraph two. The position of cell divisions is not uniform (e.g., Maciejowski et al.). Please restate.

Restated to read “cell divisions are distributed throughout the progenitor zone”.

8) Subsection “FBF-1 and FBF-2 differentially regulate mRNA abundance of target genes controlling proliferation and differentiation”. Early adult glp-1(ar202) shifted to 25º as L1 does not contain "only mitotic cells" (Pepper et al., 2003). Please restate.

We restated to read “containing a large number of mitotic cells”. A highly penetrant phenotype of the early adult glp-1(ar202gf) is proximal proliferation rather than a fully-tumorous germline (Pepper et al., 2003). However, depending on the rearing conditions, late-onset fully-tumorous phenotype is often observed (Pepper et al., 2003). In our experiments, we often observe fully tumorous phenotype with only a minor contribution of gametogenesis at the restrictive temperature. We now document this approach in more detail in Materials and methods, and include the REC-8/pH3 staining images in Figure 5—figure supplement 1A.

9) I found the naming of the vr mutants and swaps made this section hard to follow. Perhaps you could name them based on the donor and target proteins?

We have changed the names of the VR mutants as suggested to indicate the target protein followed by the donor protein and variable region number in parentheses; for example, FBF-1(FBF-2vr4).

10) Discussion paragraph two. This point is subtle and a bit confusing. Perhaps clarify that you mean the proportions of cycling-competent versus non-cycling-competent pre-meiotic S that normally contribute to the PZ?

We have re-written this statement as suggested.

11) In the same paragraph add "specific", "no specific molecular markers…"

Changed as suggested.

Reviewer #2:[…] However, the data provided by the Wang et al. suffers from improper statistical analysis, imprecise interpretation of result and needs additional data to support the claims in the manuscript.1) FBF-1 and FBF-2 differentially modulate proliferation and meiotic entry of *C. elegans* germline SPCs.The authors have found that fbf mutants have differential mitotic index and concluded that they have different rate of proliferation. For the fbf-2 mutant, a longer cell cycle is supported by slightly longer G2 phase. However, for the fbf-1 mutant G2 length is the same as wild type, which is inconsitent with the higher mitotic index (Lara-Gonzalez et al., 2019). Higher M-phase index could instead be due to cells spending more time in M-phase (which is possible due to overexpression of cell cycle regulators such as cyclin B proteins, which is downregulated to trigger M-phase exit). Additional data will need to be presented before making the conclusion that the fbf-1 mutant has a shorter cell cycle. Determining the S-phase index could provide support for a shorter cell cycle. The S-phase index data could also bolster the point that fbf-2 has a longer cell cycle, as the G2 lengthening is very modest and thus unlikely to explain the decreased meiotic output.Importantly, the result that meiotic output of fbf-1 and fbf-2 mutants is higher and lower, respectively, than wild type is sufficient to explain the differences in the SPC zone length and FBF proteins role in meiotic entry. However, the data suffered from low sample size.

We apologize for being unclear: our conclusion from cell cycle analysis in the wild type and *fbf* mutants was that wild type and *fbf-1* have similar cell cycle lengths, while *fbf-2* mutant has a longer cell cycle length. We now emphasize this conclusion in the Results section immediately after reporting the mitotic index increase in *fbf-1* by telling that the hypothesis of faster cell cycle in *fbf-1* mutant was later rejected to avoid confusion.

We added an alternative interpretation that *fbf-1* cells spend longer time in M-phase in Discussion.

We now report new experiments to provide additional analysis of cell cycle in the three genotypes, including determination of the S-phase index in the adult and documenting accumulation of germline stem cells during larval development (Figure 1E, Figure 1—figure supplement 1D, E, F; see also our response to Essential Revisions).

Finally, we respectfully disagree with regard to the sample size used for meiotic entry rate determination. The rate of meiotic entry was estimated in five independent experiments, each using 5-7 germlines per strain per time point. This comes up to 89-94 total germlines scored per strain, which is larger than the sample size utilized in previous reputable publications (eg., Kocsisova, Kornfeld and Schedl, 2019 Development score 49-57 total germlines per strain). We now report the total number of germlines analyzed per strain in the legends of Figure 1 and Figure 1—figure supplement 1 to help communicate this point.

2) FBF-1 and FBF-2 differentially regulate mRNA abundance of target genes controlling proliferation and differentiation.The authors hypothesized that "FBF-1 and FBF-2 might be acting differentially on the same set of mRNAs". They chose a set of mRNA targets of FBF (3 meiotic genes and 4 cell cycle regulators). Based on qPCR in fbf mutants and wild type (all in glp-1(gf) background, Figure 1G), authors argue that FBF-1 destabilize these target mRNAs, while FBF-2 promotes their accumulation. This data lays the foundation for their subsequent analysis. However, the data need improvement in several ways:i) The authors have chosen linear trend analysis to show that, for all the mRNA tested, levels are higher in fbf-1(-) > wild type > fbf-2. While trend looks to be true, but given that control, tbb-2 mRNA levels are reduced by almost half in fbf-2 mutant compared to wild type, with a possible exception cyb-2.2, none of the mRNAs tested can be different in fbf-2 mutant compared to wild type. Based on the graph one can only argue that FBF-1 destabilize mRNA levels but FBF-2 does not have any appreciable effect.

As suggested, we now replaced linear trend analysis with comparisons between the wild type (*glp-1gf*) and *fbf-2* mutant backgrounds. Additionally, we have replaced *tbb-2* with a different housekeeping gene, *unc-54* (myosin heavy chain). These new data are reported in Figure 5—figure supplement 1B.

ii) The statistical labels in Figure 1G is inconsistent with the statement made in the result. The authors claimed that other than cyb-1, all the other targets mRNA expression are increased in fbf-1(-), but reduced in fbf-2(-). While in Figure 1G, the statistical significance label for htp-2 and cyb-1 are EXACTLY the same. In addition, the authors are encouraged to explain why one-way ANOVA was chosen for their statistical analysis in the Materials and methods section. A pairwise comparison between wild type and each fbf single mutant should be performed for all the mRNAs tested; if there is a significant difference then one can make a conclusion.

ANOVA analysis is required to determine whether there is statistically significant difference among any of the data sets in groups of 3 or more sets, but does not indicate which data sets are different from each other. Therefore, it returns same significance for a group where *fbf-1(-)* is significantly different from both wt and *fbf-2(lf)* and a group where *fbf-2(-)* is different from *fbf-1(-)* and wt. Following detection of significant difference by ANOVA, a post-hoc test is used to determine which specific samples are different from each other. In response to the reviewer’s suggestion, we now follow ANOVA analysis by a Student’s t-test to compare the abundance of FBF target mRNAs in *fbf-2; glp-1* mutants to those of *glp-1* single mutants instead of linear trend analysis.

iii) The authors indicate that they are studying these targets' regulation in germline tumor, however the RNA is from whole worms. The authors are encouraged to explain their approach.

Thank you for this suggestion. Materials and methods section now includes the explanation that FBF targets investigated in this paper are expressed in the germline and not in the somatic cells of the adult worms. Therefore, RNA abundance in whole worm lysates reports on germline-specific expression of these mRNAs.

3) Repression of cyclin B by FBF limits accumulation of germline SPCs.The authors have mutated all three FBEs in cyb-2.1 and have successfully shown that CYB-2.1 protein levels have become higher in immunoblot. They also show that this mutation causes an increase of SPC size in fbf-2 mutant and this can be reverted by depleting cyb-2.1. Based on this the authors conclude that "levels of B-type cyclins limit SPC proliferation rate in fbf-2(lf) and disruption of FBF-1-mediated repression of a single cyclin B gene is sufficient to affect the size of germline SPC zone".There are multiple issues with this result:i) *C. elegans* has ~12 puf genes (Prasad et al., 2016) and most of them have not been fully characterized; thus it is possible that mutated FBEs might be recognized by one or more of these puf proteins rather than FBF. Immunostaining and quantification of CYB-2.1 in extruded germlines of the fbf-1 mutant and the fbf-2 mutant would be required to claim that there is reciprocal regulation by the FBF proteins, rather than other puf proteins.

We have restated our conclusions to reflect that FBEs in *cyb-2.1* 3’UTR might mediate repression through recruitment of other PUF proteins expressed in progenitor cells, although the documented differences between PUF proteins’ preferred binding sites (Bernstein et al., 2005; Opperman et al., 2005; Koh et al., 2009) would suggest that FBEs are likely recognized by FBFs rather than other PUFs expressed in the SPCs.

ii) Mutating FBEs in cyb-2.1, results in its higher expression. Rationale for checking the effect of its overexpression in fbf-2(-) background, instead of wild type, is not clear. Since there are no FBEs in cyb-2.1fbm, its effects are expected to be fbf-agnostic.

We now emphasize the rationale that the *fbf-2(-)* with its slow cell cycle rate provides a sensitized background to detect the effects of cell cycle regulator overexpression, since it is unclear whether cell cycle rate can be accelerated beyond that of the wild type.

iii) Subsection “Repression of cyclin B by FBF limits accumulation of germline SPCs” paragraph three authors have concluded that increase in SPC zone size in fbf-2; cyb-2.1fbm vs fbf-2 is due to increased proliferation and also the meiotic entry rate is unchanged between two mutants. They have not provided any evidence for either of these two possibilities, i.e. proliferation rate has increased in fbf-2; cyb-2.1fbm and meiotic entry rate is same in fbf-2; cyb-2.1fbm as compared to fbf-2.

Thank you for this suggestion. New experiments in response to these suggestions are summarized in Essential Revisions. As suggested, we now include the estimation of stem cell proliferation rate comparing *fbf-2(lf)* with the strains expressing wild type or mutant *3xflag::cyb-2.1* transgenes. These new experiments suggest that cell cycle rate is accelerated in *fbf-2(lf); 3xflag::cyb-2.1fbm* compared to *fbf-2(lf)* (Figure 2D). By contrast, meiotic entry rate of *fbf-2(lf)* is not affected by expression of either cyb-2.1 transgene (Figure 2E).

4) FBF-1 promotes deadenylation of its target mRNA.The authors moved back to cyb-2.1 to show that poly A tails are affected in fbf mutants without giving any rationale. The assay should be performed on a few more genes if any generalized statement statement is made, otherwise it must be restricted to cyb-2.1. Also wild type (glp-1(gf)) is missing in the assay, which makes it hard to interpret the data. Is FBF-1 alone or both FBFs responsible for deadenylation of cyb-2.1 mRNA or if FBF-1 and FBF-2 have opposing effect on cyb-2.1 mRNA?

As suggested, we now include the wild type (*glp-1(gf)*) as a control for PAT-PCR assay (Figure 5), and find that FBF-1 and FBF-2 have opposing effects on *cyb-2.1* poly(A) tail. Furthermore, we have included PAT-PCR assay for an additional FBF target, *htp-1*. Our choice of transcripts for analysis represents both cell cycle-related transcripts and differentiation-related transcripts. We focused on the targets for which we could obtain reliable and robust amplification of poly(A) tails.

5) While the authors present some solid lines of evidence that FBF-1 and FBF-2 are different (meiotic entry rate, FBF-1 acts with CCR-4/NOT to de-adenylated cyb-2.1 and there are specific sequences in FBF-2 that block this interaction), they are clearly redundant with each other for the stem cell fate and repression of htp-2. Thus, Figure 8 would be more useful to the community if their redundant function was also included.

The revised model (Figure 8) now emphasizes that both FBFs repress the bound target mRNAs and both FBFs can maintain SPCs on their own (thus highlighting their redundant function).

6) FBF-1 and FBF-2 do not explain the spatial control of expression in the SPC zone and thus balance between proliferation and differentation, which is through glp-1 targets lst-1 and sygl-1. It is surprising that there is no mention of lst-1 and sygl-1-1 in the Discussion, particularly since they are co-factors of FBF-1 and FBF-2 (Haupt et al., 2019, Shin et al., 2017).

We now include LST-1 and SYGL-1 in the Discussion. We have not investigated whether the changes in SPC zone size in *fbf* mutants are associated with altered expression of LST-1 or SYGL-1 (that are curiously themselves FBF targets), and it would be an interesting new direction to explore.

Reviewer #3:[…] My critique of this manuscript can be broken into two categories: (1) it would benefit from some changes in writing to increase clarity and accessibility for non-experts and (2) there are a few questions that I had regarding experimental issues, some related to interpretation, which could be addressed in the text (points D, and F-H below) and some related to additional controls/experiments that would substantiate the authors' main conclusions (points A-C, E, and F). The suggested experiments are not substantially outside of those that the authors have already conducted and I think that most of them (enough to satisfy the majority of my major comments) could be performed within a reasonable time frame.Major comments:1) Writing/interpretation:As someone who works in the germline, but not on this pathway in particular, I found that this paper was often hard to digest. It would be very helpful if the authors could make sure that their rationale for each experiment and its interpretation is accessible to a reader not intimately familiar with the field. For example, the difference in proliferative zone (PZ) size in fbf-1(lf) and fbf-2(lf), I think pointing out here how a change in PZ size shows a shift in the balance between proliferation and differentiation, as if both were slowed down or sped up in complete coordination, the size of PZ would likely be constant. In addition, explaining why experiments were designed in a certain way and why the (perhaps) easier experiment is not feasible would be very helpful.

We have revised to reflect that the changes in SPC zone size in *fbf-1* and *fbf-2* mutants suggest that each individual FBF protein does not perfectly balance the cell division and meiotic entry. Rather, antagonistic FBF activities combine to simultaneously modulate stem cell division with the rate of meiotic entry. Although our model might be oversimplified, it provides a framework for future exploration of factors controlling SPC dynamics.

Additionally, we have clarified the rationales for the reported experiments.

I also believe that the authors need to revise their language in several instances to avoid conflating two different questions/fields. For example, I found the use of "differentiation demands" confusing, because, for me, "differentiation demands" implies reproductive demand, i.e. the need for mature gametes. In both the *C. elegans* germline (e.g. Narbonne, et al., 2015) and in other stem cell populations (e.g. Hsu, Li and Fuchs, 2014), signaling from differentiated/differentiating progeny to their stem cell parents plays a significant role in regulating stem cell proliferation, such that proliferative output matches the demand for differentiated progeny. I do not think that the authors are addressing this type of "feedback" signaling – i.e. they do not show that the rate of meiotic entry regulates the rate of proliferation, but rather that these two things are coordinated. I think that it is important to clarify their questiion to avoid confusion and/or erroneous expectations from people, such as myself, who are more used to thinking about feedback mechanisms. There are several instances of this throughout, including: "Complementary activities of FBF-1 and FBF-2 combine to fine tune SPC proliferation and differentiation to respond to proliferative demands of the tissue."

Thank you for this suggestion. We have clarified our writing to convey that the focus here is on coordinate regulation of cell division rate and meiotic entry.

2) Experimental issues:A) Figure 1. The cell cycle phenotype of fbf-1(lf) is puzzling. Since there are no quiescent cells in the germline, an increase in mitotic index (MI) ought to reflect either a faster passage through the mitotic cell cycle or a longer duration of mitosis. The authors report no change in the length of G2, which leads me to wonder whether S phase is accelerated or mitosis is delayed (I believe there are assays to assess both (S phase, Fox and Schedl, 2011) and M phase (Gerhold et al., 2015)). This result is also odd in light of their observation that the transcript levels of most of the Cyclin Bs (except for CYB-1), are highly elevated – i.e. why is the duration of G2 unaffected despite higher levels? The authors explanation in the discussion that a higher MI could be due to a smaller proliferative population, due to accelerated meiotic entry, would predict (I think) that the stem cell pool should eventually be lost – i.e. in older fbf-1(lf) adults, do they see a progressive loss of proliferative cells? As this interpretation (i.e. FBF-2, rather than FBF-1, being primarily responsible for cell cycle regulation) is key to their model figure at the end, it bears a closer look.

Thank you for these suggestions. We addressed many of these in response to the Essential Revisions. Briefly, we determined the S-phase index of the adult germlines, and find that S phase is not accelerated in *fbf-1(lf)*. We were not able to determine the duration of M-phase since we lack the required live imaging microfluidic set up. Additionally, an assay measuring germline stem cell accumulation during larval development suggested that overall division rate of *fbf-1(lf)* and wild type germ cells is the same.

Furthermore, we clarified the potential explanation for higher mitotic index: the total SPC cell count includes non-cycling cells in meiotic S-phase (they cannot be excluded because the field lacks specific molecular markers for this developmental stage). Accelerated meiotic entry in *fbf-1(lf)* might selectively reduce this non-cycling cell population due to their faster transit through the meiotic S-phase. This would lower overall SPC cell counts (without reducing the number of proliferative cells), and thus lead to a higher MI.

B) Also Figure 1. I am unsure as to how to interpret the RT-qPCR data. The authors state that their negative control (tbb-2) is unaffected, yet in fbf-2(lf) transcript levels appear to be at least a strongly depressed as those of the genes that they argue are maintained by FBF-2. It is possible that I am missing something here, but it would be good if the authors could address/explain this, as it is visually striking/distracting in their data/graph and if it is accurate, i.e. that in fbf-2(lf) transcript levels are generally lower, it is problematic for the interpretation that FBF-1 and FBF-2 have reciprocal effects on shared target transcripts. It is also not entirely clear to me what the linear trend test adds to the data interpretation, some clarification/discussion would be helpful.

We addressed these suggestions in the Essential Revisions. Briefly: First, the transcript levels in Figure 1G (now Figure 5—figure supplement 1B) are normalized to actin (*act-1*), so a decrease in FBF target transcripts in *fbf-2(lf)* does not reflect generally lower transcript levels. Additionally, we have tested other housekeeping mRNAs to provide an alternative negative control. We now replace *tbb-2* with *unc-54*, which abundance changes in the opposite direction to that of FBF targets. Second, as suggested, we now replaced linear trend analysis with comparisons between the wild type (*glp-1gf*) and *fbf-2* mutant backgrounds.

C) In Figure 2, the authors address the hypothesis that slow G2 and low MI in fbf-2(lf) is due to FBF-1-mediated translational repression of cyb-2.1. The authors state that protein levels for CYB-2.1 with mutated FBF binding elements (FBEs) are elevated, but they cannot detect protein levels for the WT version, which could imply that it is not expressed at all. It would be helpful if the authors could should that both transgenes are expressed by looking at transcript levels. I also wonder if this result predicts that in wildtype germlines, expression of cyb-2.1 with mutated FBEs, but not wild-type cyb-2.1 would affect cell cycle progression? Finally, this hypothesis also seems to predict that a partial RNAi of FBF-1 in a fbf-2(lf) background, should rescue, at least to some extent, the fbf-2(lf) phenotype. If this experiment is technically feasible, it would greatly strength the authors argument.

Thank you for these excellent suggestions!

1) We have generated a new *cyb-2.1wt* transgene that we can now detect both at the transcript and at the protein level, and report these new data in Figure 2 (Figure 2B, C). Crossing the new *cyb-2.1wt* into *fbf-2(lf)* mutant did not affect the stem cell dynamics (Figure 2D-G).

2) As mentioned in the response to reviewer 2 (3ii), we now emphasize the rationale that the *fbf-2(-)* with its slow cell cycle rate provides a sensitized background to detect the effects of cyclin overexpression.

3) We have not been able to partially knock down *fbf-1* to address whether this would be able to rescue *fbf-2(lf)*.

D) In Figure 3, the authors use a translational reporter for the htp-2 3'UTR, which their RT-qPCR results suggest should be derepressed in fbf-1(lf), but they do not seem to see this and I am unsure as to why not? In addition, their results strongly suggest that the change in PZ size in fbf-2(lf) requires CCR4-NOT complex members, which supports the hypothesis that FBF-1 activity with CCR4-NOT is leading to transcript loss. However, if FBF-1 acts with CCR4-NOT to regulate transcript levels, shouldn't RNAi of CCR4-NOT complex members phenocopy fbf-1(lf)? It would be helpful if the authors could address these questions. This may be another instance of where the data/results are not fully accessible to a non-expert.

We now explain that FBF targets are not derepressed in *fbf-1* mutant even though their mRNAs are accumulating in cytoplasmic foci (Merritt and Seydoux, 2010; Voronina et al., 2012) since FBF-2 on its own is capable of translational repression. However, FBF-2-mediated repression might be less effective since partial deprepression of several targets including *gld-1* has been previously observed in *fbf-1(lf)* mutants (Crittenden et al., 2002, Brenner and Schedl, 2016). To test whether *fbf-1* requires CCR4-NOT for function, we assess whether *fbf-2(lf)* after CCR4-NOT subunit RNAi show the same range of phenotypes as *fbf-1(lf) fbf-2(lf)*. These phenotypes include derepression of the *htp-2* 3’UTR reporter, sterility, and failure to initiate oogenesis. This reasoning is now included in the Results section.

E) In Figure 5, the authors quantify the polyA tail length of cyb-2.1 and show that it is shorter in fbf-2(lf), suggesting that it is being deadenylated, presumably by FBF-1/CCR4-NOT. It would strengthen the authors conclusion substantially, if they could show that the change in polyA tail length was rescued by depletion of CCR4-NOT complex members. It is also unclear to me why the authors compared fbf-1(lf) to fbf-2(lf) rather than to wildtype? I also wonder whether tbb-2 is the best control as the detection levels seem very low which could make slight changes in peak detection challenging.

As suggested, we have added the wildtype control (*glp-1(gf)*) to the polyA tail length assay (Essential Revisions), and now use *unc-54* myosin instead of *tbb-2* as the control. These changes resulted in revised Figure 5. While testing whether changes in polyA tail length in *fbf-2(lf)* are dependent on CCR4-NOT would provide additional evidence supporting FBF-1/CCR4-NOT regulatory association, we did not have the time to perform the assay as large-scale culture of *fbf-2(lf); glp-1(gf)* nematodes on RNAi is challenging. Nevertheless, multiple lines of evidence presented in the paper strongly support this conclusion.

F) Figure 6: I think that this a key place where the authors interpretation could be clarified somewhat. I believe that in Figure 6, the fact that FBF-2 without variable regions (VRs) 1, 2 and 4 cannot fully rescue the fbf-2(lf) phenotype in either a fbf-2(lf) or a fbf-1(lf) fbf-2(lf) background suggests that either this version of FBF-2 is only partially functional or it is acting like FBF-1. The authors then provide support for the latter interpretation by showing that the activity of FBF-2(vrm) requires CCR4-NOT and can also partially rescue fbf-1(lf). This last piece of data is in Figure 6—figure supplement 1, but, in my opinion, should be moved into the main figure panel. This also suggests that FBF-1 may be able to act with CCR4-NOT without any of its own variable regions, which (I think?) is somewhat surprising. The authors could test this by using a similar approach.

As suggested, we have moved the *fbf-1(lf)* rescue from Figure 6—figure supplement 1B to the main Figure 6. We agree that our data suggest that FBF-1’s variable regions flanking the RNA-binding domain are permissive for acting with CCR4-NOT. One simple possibility is that VR4 of FBF-2 includes a 16-amino acid C-terminal extension that might be sterically interfering with the recruitment of CCR4-NOT. However, rigorously testing this and other possible determinants allowing FBF-1/CCR4-NOT cooperation is beyond the scope of the present work.

G) Figure 7: For the domain swapping experiments here, I believe the main conclusions are that VR3 (in the RNA-binding domain) is not required for FBF-1 activity, and that VR4 from FBF-2 is sufficient to prevent FBF-1 from acting with CCR4-NOT to regulate polyadenylation, as FBF-1 with FBF-2's VR4 cannot rescue fbf-1(lf) in a fbf-1(lf) or a fbf-1(lf), fbf-2(lf) background, while FBF-1 with FBF-2's R3 can. However, what I do not understand is why FBF-1 with FBF-2's VR4 rescues fbf-2(lf) (Figure 7—figure supplement 1)? Does this mean that it is acting as a dominant negative, since presumably wild-type FBF-1 is still present in this background?

We propose that FBFs bind to the same mRNAs, and either compete for the same binding sites or simultaneously bind to the transcripts that contain multiple FBEs. Either mechanism would explain how change-of-function mutants such as FBF-2vrm and FBF-1(FBF-2vr4) are able to rescue the loss-of-function of the non-cognate *fbf*. We do not think that change-of-function mutants are actually dominant-negative (i.e., poison the function of the original FBF), because in that case a combination of FBF-1(FBF-2vr4) with the wild type *fbf-1* in the *fbf-2(lf)* background would disrupt the activity of the sole remaining wt FBF-1 and produce a phenotype similar to a loss of function of both *fbfs* (loss of SPCs in the adult).

H) Figure 8: The model figure as currently drawn doesn't really help me understand the authors' mains conclusions/findings. It would be great if this could be re-worked to reflect the effects of FBF-1 and FBF-2 on the stem cell population (i.e. perhaps in relation to PZ size which is their phenotypic readout for most experiments) and how these activities lead to a balance between stem cell maintenance and differentiation.

The model in Figure 8 now depicts that FBF effects on the extent of the mitotic zone result from their regulation of stem cell dynamics (how fast the stem cells are able to divide and the rate of meiotic entry).